# The ESRP1-GPR137 axis contributes to intestinal pathogenesis

Lukas Franz Mager[1,2], Viktor Hendrik Koelzer[1], Regula Stuber[1], Lester Thoo[1,2], Irene Keller[3,4], Ivonne Koeck[1,2,3], Maya Langenegger[1], Cedric Simillion[3,4], Simona P Pfister[2,5], Martin Faderl[1,2], Vera Genitsch[1], Irina Tcymbarevich[6], Pascal Juillerat[7], Xiaohong Li[8], Yu Xia[9], Eva Karamitopoulou[1], Ruth Lyck[10], Inti Zlobec[1], Siegfried Hapfelmeier[5], Rémy Bruggmann[4], Kathy D McCoy[3†], Andrew J Macpherson[3,7], Christoph Müller[1], Bruce Beutler[8], Philippe Krebs[1]*

[1]Institute of Pathology, University of Bern, Bern, Switzerland; [2]Graduate School for Cellular and Biomedical Sciences, University of Bern, Bern, Switzerland; [3]Department of BioMedical Research, University of Bern, Bern, Switzerland; [4]Interfaculty Bioinformatics Unit and Swiss Institute of Bioinformatics, University of Bern, Bern, Switzerland; [5]Institute for Infectious Diseases, University of Bern, Bern, Switzerland; [6]Division of Gastroenterology and Hepatology, University Hospital Zurich, Zurich, Switzerland; [7]Department of Gastroenterology, Inselspital, Bern University Hospital, University of Bern, Bern, Switzerland; [8]Center for Genetics of Host Defense, University of Texas Southwestern Medical Center, Dallas, United States; [9]Department of Genetics, The Scripps Research Institute, La Jolla, United States; [10]Theodor Kocher Institute, University of Bern, Bern, Switzerland

*For correspondence:
philippe.krebs@pathology.unibe.ch

Present address: †Department of Physiology and Pharmacology, Synder Institute for Chronic Diseases, Calgary, Canada

**Abstract** Aberrant alternative pre-mRNA splicing (AS) events have been associated with several disorders. However, it is unclear whether deregulated AS directly contributes to disease. Here, we reveal a critical role of the AS regulator epithelial splicing regulator protein 1 (ESRP1) for intestinal homeostasis and pathogenesis. In mice, reduced ESRP1 function leads to impaired intestinal barrier integrity, increased susceptibility to colitis and altered colorectal cancer (CRC) development. Mechanistically, these defects are produced in part by modified expression of ESRP1-specific *Gpr137* isoforms differently activating the Wnt pathway. In humans, *ESRP1* is downregulated in inflamed biopsies from inflammatory bowel disease patients. ESRP1 loss is an adverse prognostic factor in CRC. Furthermore, generation of *ESRP1*-dependent *GPR137* isoforms is altered in CRC and expression of a specific *GPR137* isoform predicts CRC patient survival. These findings indicate a central role of ESRP1-regulated AS for intestinal barrier integrity. Alterations in ESRP1 function or expression contribute to intestinal pathology.

DOI: https://doi.org/10.7554/eLife.28366.001

## Introduction

The single-layered intestinal epithelium provides an important physical barrier that critically contributes to intestinal homeostasis (*Peterson and Artis, 2014*). Dysfunction of intestinal epithelial cells (IECs) leading to increased epithelial permeability is associated with intestinal diseases such as inflammatory bowel disease (IBD) and colorectal cancer (CRC) (*Van der Sluis et al., 2006*; *Schmitz et al., 1999*; *Grivennikov et al., 2012*). IBD is related to polymorphisms in various IBD susceptibility genes (*Lees et al., 2011*; *Van Limbergen et al., 2014*), and numerous genetic alterations in key cellular pathways that underlie CRC have been identified (*Fearon, 2011*). However, the role

**eLife digest** The lining of the intestine is just one cell thick, and yet it can act as an effective barrier between the inside of the body and the contents of the digestive system. This lining is often disturbed during bowel cancer, inflammatory bowel disease and other intestinal diseases, causing the barrier to fail and the gut to become leaky. These diseases often reduce patient life expectancy and quality of life.

Intestinal epithelial cells make up the lining of the intestine and the normal activities of these cells are often disturbed during intestinal disease. In the intestine, a protein called ESRP1 is only found in epithelial cells, but its role in maintaining a healthy intestinal lining was not clear. Here, Mager et al. studied the intestines of mice that had been genetically engineered to produce a form of ESRP1 that is less active than normal.

The experiments show that lower levels of ESRP1 activity leads to a broken intestinal barrier. The genetically engineered mice were more likely to develop inflammatory bowel disease and more aggressive forms of cancer. ESRP1 controls a gene that encodes another protein called GPR137, which helps to relay signals to the epithelial cells. Lower levels of ESRP1 resulted in a longer form of the GPR137 protein being produced. This in turn affected the protein's signaling role and disturbed the activities of intestinal epithelial cells.

Further experiments on biopsies taken from patients with inflammatory bowel disease or colorectal cancer revealed that these patients had lower levels of ESRP1 compared to healthy individuals. Furthermore, low levels of ESRP1 and increased levels of the long version of GPR137 were associated with poorer outcomes for cancer patients. Together, these findings may help us to better understand how the intestinal barrier fails in mice and humans and could lead to new ways to monitor and treat intestinal disease.

DOI: https://doi.org/10.7554/eLife.28366.002

of post-transcriptional modifications in the regulation of IBD and CRC development is still poorly understood.

Alternative splicing of pre-mRNAs (AS) is a common posttranscriptional modification that is estimated to occur in 92–94% of human genes (*Wang et al., 2008*; *Pan et al., 2008*). AS permits generation of protein isoforms with related, distinct or sometimes even opposing functions (*Vorlová et al., 2011*). Moreover, certain isoforms influence cancer progression (*Brown et al., 2011*) or are associated with autoimmune and inflammatory disorders (*Ueda et al., 2003*; *Laitinen et al., 2004*). While alterations of mRNA splicing have been reported in human colorectal cancer (CRC) (*Freund et al., 2015*; *Zhou et al., 2014*) and IBD (*Häsler et al., 2011*; *Mailer et al., 2015*), the consequences of deregulated AS for intestinal homeostasis and disease development are largely unexplored.

Epithelial splicing regulatory protein 1 (*ESRP1*), which is exclusively expressed in epithelial cells, was identified in a cDNA expression screen for factors that promote the epithelial pattern of *FGFR2* (Fibroblast Growth Factor Receptor 2) splicing. *ESRP1* was also found to regulate the AS of *CD44* and other genes (*Warzecha et al., 2009*). ESRP1 is negatively regulated by mesenchymal transcription factors such as SNAIL, ZEB1 and ZEB2 (*Preca et al., 2015*; *Saitoh, 2015*; *Reinke et al., 2012*). Recently, *ESRP1* has been reported to act as a tumor suppressor by negatively regulating epithelial-to-mesenchymal transition (EMT) and the metastatic potential of human breast cancer cell lines, via the splicing of different isoforms of *CD44* or *EXO70* (*Lu et al., 2013*). ESRP1 can suppress cancer cell motility in head and neck carcinoma cell lines (*Ishii et al., 2014*) and ESRP1 protein expression is a favorable prognostic factor in pancreatic cancer (*Ueda et al., 2014*). However, ESRP1 may also promote lung metastasis of orthotopically transplanted breast cancer cells by generating *CD44* isoforms independently of EMT (*Yae et al., 2012*). Yet, the contribution of ESRP1 to intestinal integrity and function is poorly investigated. Importantly, the lack of viable animal models so far has precluded analyzing the role of ESRP1-mediated AS for intestinal disease in vivo.

Here, we used a novel mutant allele of *Esrp1* called *Triaka* to investigate the function of ESRP1 in the intestine. We found that *Esrp1^Triaka* (later referred to as *Triaka*) leads to reduced ESRP1 function causing distinct alterations in the mRNA splicing pattern in colonic IECs (cIECs) of *Triaka* compared

with wild-type (WT) animals. These changes in several transcript isoforms do not alter intestinal histo-morphology, yet they are associated with increased intestinal permeability in *Triaka* mice. In addition, *Triaka* mice show alterations in distinct models of intestinal disease. Mechanistically, this phenotype can be ascribed, in part, to changes in the relative frequency of specific *Gpr137* splicing isoforms in *Triaka* cIECs. This affects the survival and function of cIEC by altering the Wnt signaling pathway in *Triaka* mice. In humans, *ESRP1* transcript levels are downregulated in inflamed compared with non-inflamed biopsies from IBD patients. Furthermore, ESRP1 expression is gradually lost during the adenoma to carcinoma sequence in CRC, and loss of ESRP1 protein expression in CRC tumors negatively correlates with patient survival. Moreover, the ratio of specific *GPR137* isoforms is different in tumor versus normal intestinal tissue, and expression of a specific *ESRP1*-dependent *GPR137* isoform predicts CRC patient survival.

Together, these data indicate an important role for ESRP1 in intestinal disease in humans and mice.

## Results

### *Esrp1^Triaka* reduces mRNA splicing function

To study the role of ESRP1 in the intestinal epithelium, we used a novel mutant allele of *Esrp1* called *Triaka* that was identified in an *N*-ethyl-*N*-nitrosourea genetic screen. The *Triaka* point mutation results in a methionine-to-valine substitution at the amino acid position 161 (M161V) of ESRP1, a residue conserved in several species (*Figure 1—figure supplement 1*). *Triaka* animals develop overtly normal (see also Materials and methods), which allows investigation of the physiological role of *Esrp1* in adults. This is in contrast to the recently published *Esrp1^-/-* or skin epithelial-specific *Esrp1*-deficient animals that are neonatal lethal (*Bebee et al., 2015*). We first utilized a previously described in vitro reporter system to characterize the effect of *Esrp1^Triaka* on the splicing of known ESRP1-regulated transcripts (*Brown et al., 2011*). Using this system based on luciferase expression, we found that WT ESRP1 protein led to a 2.4 and 10.9 fold increase of *Cd44* variable exon v5 (*Cd44v5*) and *Fgfr2* variable exon IIIb (*Fgfr2*-IIIb) inclusion, respectively, compared to control. This was in line with earlier reports (*Brown et al., 2011*; *Warzecha et al., 2009*). ESRP1^Triaka however showed reduced levels of *Cd44v5* and *Fgfr2*-IIIb inclusion, with 1.8 and 3.3 fold induction, respectively (*Figure 1A and B*). This variation in the extent of in vitro splicing of *Cd44* versus *Fgfr2* by ESRP1^Triaka likely related to the fact that *Esrp1*-regulated splicing events show distinct sensitivity to *Esrp1* loss (*Warzecha et al., 2009*; *Bebee et al., 2015*).

We then transduced CMT-93 cells with inducible lentiviral encoding *Esrp1*^WT or *Esrp1^Triaka* to assess the effect of the mutation on epithelial cell function. At the same level of overexpression as *Esrp1*^WT, *Esrp1^Triaka* led to diminished inclusion of *Cd44v4/5* in CMT-93 cells and reduced cell proliferation (*Figure 1C–E*). Of note, *Esrp1*^WT transcript levels directly correlated with frequency of *Cd44v4/5* splicing events, suggesting a dose-dependent effect of *Esrp1* expression on *Esrp1*-mediated splicing activity (*Figure 1F*). Taken together, these in vitro data indicate that *Esrp1^Triaka* reduces ESRP1 function, and thereby decreases the proliferative capacity of epithelial cells.

### *Esrp1^Triaka* leads to altered mRNA splicing pattern in vivo

*Esrp1* has been shown to be epithelial cell-restricted and to be highly expressed in the murine large intestine (*Warzecha et al., 2009*). Thus, we next investigated the impact of *Esrp1^Triaka* on the mRNA splicing pattern of cIECs. RNA sequencing was performed on cIECs isolated from naïve mice. Computational analysis revealed 35 genes for which the relative frequency of splicing isoforms differed in *Esrp1^Triaka* versus *Esrp1*^WT cIECs (*Figure 2A and B*, *Figure 2—source data 1*). These findings were validated using quantitative reverse-transcription polymerase chain reaction (qPCR) for selected, previously reported ESRP1 target genes, including *Cd44* and *Magi1* (*Brown et al., 2011*; *Warzecha et al., 2009*; *Warzecha et al., 2010*), and for novel candidate targets, including *Uap1* and *Gpr137* (*Figure 2C*). Functional alterations induced by ESRP1^Triaka were also apparent on the protein level, as expression of CD44 variant 4 (CD44v4)-containing isoforms, but not total CD44, was reduced in colonic sections of *Triaka* versus WT mice (*Figure 2D* and *Figure 2—figure supplement 1*). Therefore, these results suggested *Esrp1^Triaka*-dependent functional changes in mutant mice.

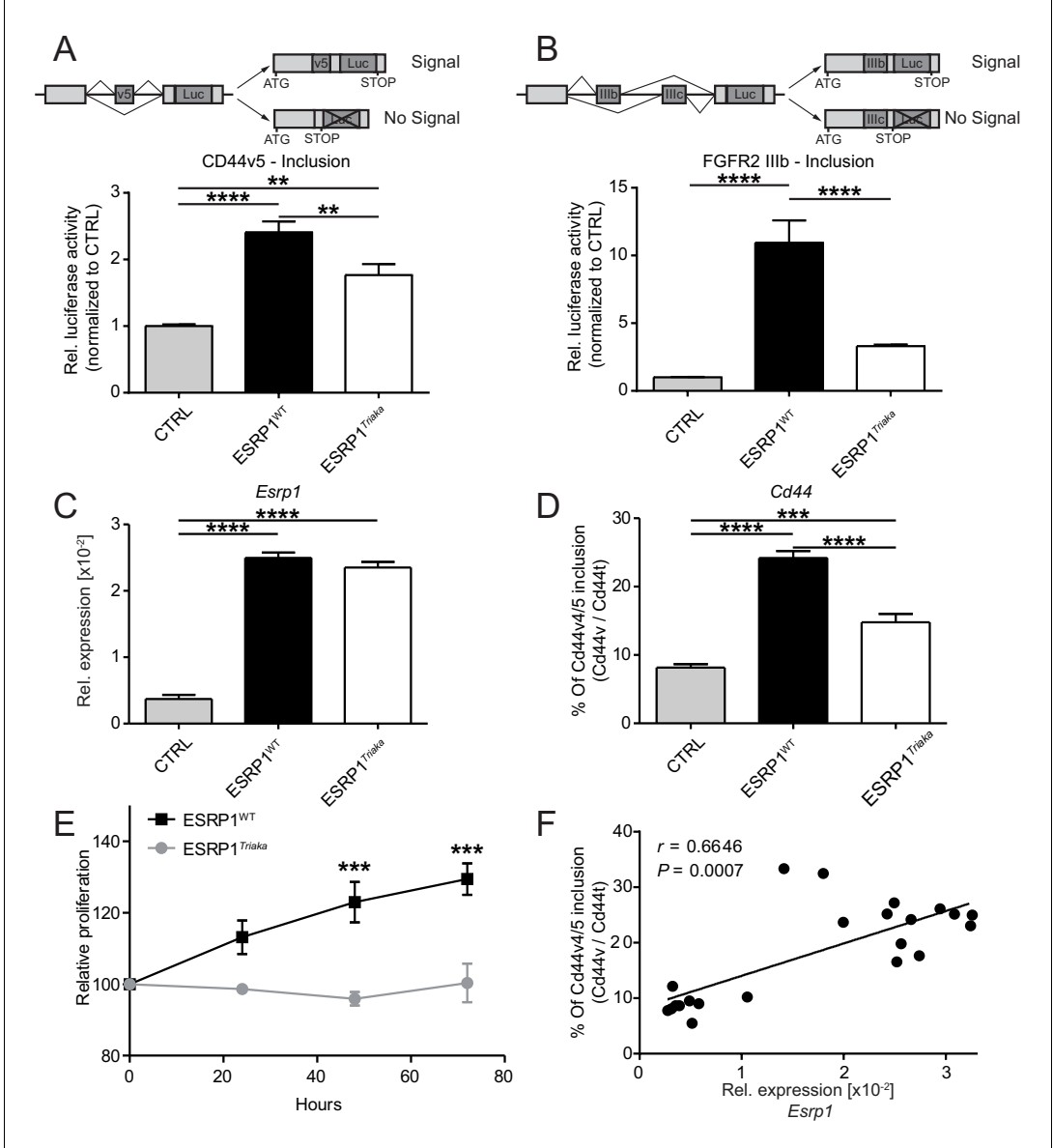

**Figure 1.** *Esrp1^Triaka* leads to altered mRNA splicing and reduced epithelial cell proliferation. To quantify exon splicing, HEK-293 cells were co-transfected with a vector encoding *Esrp1^WT* or *Esrp1^Triaka* or an empty control (CTRL) vector and with an exon trap construct containing (**A**) *Cd44* variant 5 (*Cd44v5*) or (**B**) *Ffgr2*-IIIb variable exon. Upper panels in (**A**) and (**B**) show schemes of the respective exon trap constructs. Inclusion of exon *Cd44v5* or *Ffgr2*-IIIb results in luciferase (Luc) expression. Luciferase activity normalized to control vector-transfected cells is shown in the lower panels. Relative expression of (**C**) *Esrp1* and (**D**) *Cd44v4/5* transcripts were measured in CMT-93 cells transduced with inducible vectors encoding *Esrp1^WT*, *Esrp1^Triaka*, or a control construct, after treatment with 4-hydroxytamoxifen. (**E**) Proliferation of *Esrp1^WT*- and *Esrp1^Triaka*-expressing CMT-93 cells was measured using a WST-1 assay and normalized to control vector-transduced cells. (**F**) Relative expression of *Esrp1* and *Cd44v4/5* transcripts were measured in CMT-93 cells transduced with a vector encoding *Esrp1^WT*, after induction with 4-hydroxytamoxifen, and correlated. Data represent: Pooled means ± standard error of the mean from (**A and B**) five or (**C–E**) three independent experiments performed in biological triplicates. (**F**) Means measured in technical duplicates (n = 22). Statistics: (**A–D**) One-way ANOVA with Bonferroni post-test. (**E**) Two-way ANOVA with Bonferroni post-test. (**F**) Spearman correlation. **p<0.01; ***p<0.001; ****p<0.0001.

DOI: https://doi.org/10.7554/eLife.28366.003

The following figure supplement is available for figure 1:

**Figure supplement 1.** Mapping of the *Triaka* mutation.

DOI: https://doi.org/10.7554/eLife.28366.004

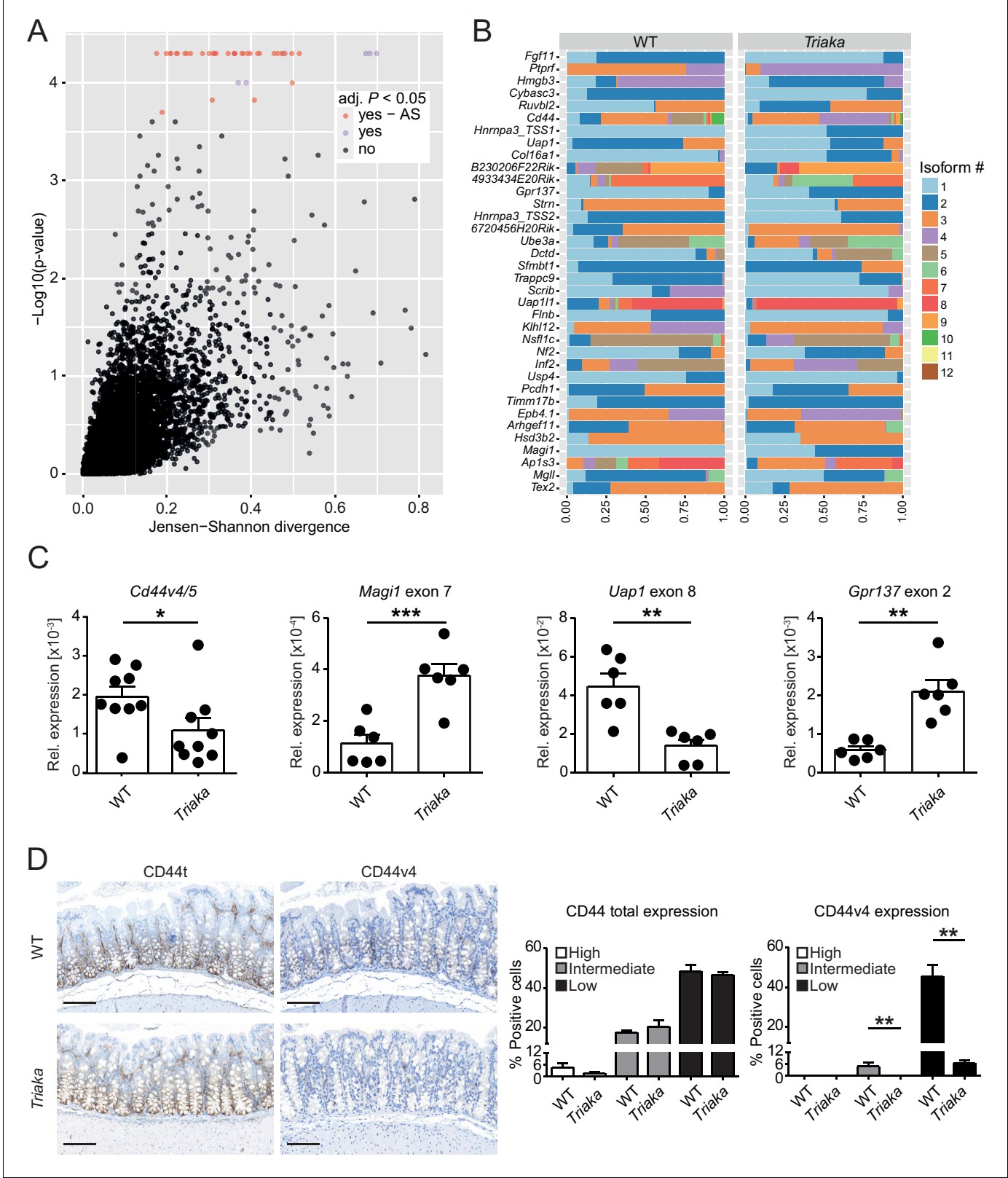

**Figure 2.** *Esrp1^Triaka* alters mRNA splicing patterns in colonic intestinal epithelial cells. (**A**) RNA sequencing analysis was performed on colonic intestinal epithelial cells (cIECs) isolated from WT and *Triaka* mice. Dot plot indicating the relative difference in isoform usage for a given transcription start site expressed as Jensen-Shannon divergence and the associated p-values. Analysis was performed through CummeRbund and FDR-adjusted p-value<0.05
*Figure 2 continued on next page*

*Figure 2 continued*

were considered significant. Red and orange dots represent genes with differences in transcript isoforms generated by bona fide alternative splicing (AS) events or by other mechanisms, respectively (*n* = 4 donor mice per group). (**B**) Panel showing the relative frequency of the different isoforms identified as AS events in (**A**) for 36 transcription start sites and from 35 genes, in WT versus *Triaka* cIECs. (**C**) Transcript levels for the indicated isoforms were measured in WT and *Triaka* cIECs using qPCR and normalized to *Gapdh* expression (*n* = 6–11 mice per group). *Cd44v4-5: Cd44* variant 4–5. (**D**) Immunohistochemistry was performed on colonic tissue of indicated mice to detect total CD44 (CD44t) or CD44v4. Representative pictures and percentage of positive cells for the indicated staining intensities are shown (*n* = 5–6 mice per group, pooled from three independent experiments). Histograms represent the mean ± standard error of the mean. Statistics: (**C**) Student's *t* test; (**D**) Mann-Whitney test. *p<0.05; **p<0.01; ***p<0.001; ****p<0.0001.

DOI: https://doi.org/10.7554/eLife.28366.005

The following source data and figure supplement are available for figure 2:

**Source data 1.** Altered ratios of transcript isoforms in *Triaka* epithelial cells.
DOI: https://doi.org/10.7554/eLife.28366.007
**Figure supplement 1.** CD44 expression in *Triaka* and WT colons.
DOI: https://doi.org/10.7554/eLife.28366.006

## *Esrp1*<sup>*Triaka*</sup> impairs the integrity of the intestinal barrier

We next compared the gene expression profiling of *Triaka* versus WT cIECs to assess potential functional effects of *Esrp1*<sup>*Triaka*</sup>. Pathway analysis of the RNA sequencing data showed that several pathways involved in cell cycle and proliferation were affected in *Triaka* cIECs (*Figure 3—figure supplement 1* and *Figure 3—source data 1*). Yet, these transcriptional changes did not overtly alter intestinal histomorphology and epithelial proliferation, as crypt depth, number of goblet cells and Ki-67 expression were similar in *Triaka* versus WT mice (*Figure 3—figure supplement 2A–F*). Furthermore, *Triaka* and WT mice showed similar colon length at steady-state (*Figure 3—figure supplement 2G*). However, E-cadherin (CDH1) surface expression was reduced on *Triaka* cIECs (*Figure 3A and B*). Given the central role of E-cadherin for epithelial cell function, we hypothesized a possible defect in intestinal epithelial barrier integrity in *Triaka* mice. Indeed, ex vivo measurement of the intestinal electrical resistance indicated an increased ion permeability of *Triaka* versus WT colonic mucosa (*Figure 3C*), although there was no such difference in the small intestine (*Figure 3—figure supplement 3A*). This was accompanied by the presence of bacterial 16S rRNA in intestinal crypts and in the inner mucus layer of *Triaka* but not WT colons (*Figure 3D* and *Table 1*). As further indirect evidence of reduced barrier integrity, we also detected systemic anti-commensal IgG1 and IgG2b antibody reactivity towards autologous intestinal bacteria in most *Triaka* mice, but rarely in control animals (*Figure 3E* and *Table 1*). However, levels of fecal albumin and lipocalin-2 – markers for intestinal lesions and inflammation – were similar in fecal pellets of *Triaka* and WT mice (*Figure 3—figure supplement 3B and C*). Furthermore, there was no difference in the resorption of macromolecules between *Triaka* and WT mice (22 ± 1 versus 24 ± 3 μg/ml serum FITC-dextran-4000 levels, respectively, p=0.6).

In summary, these data suggest that reduced *Esrp1*-dependent mRNA splicing in *Triaka* animals results in decreased integrity of the colonic epithelial barrier and intestinal penetration of bacterial products. These epithelial defects are however not sufficient to induce intestinal immunopathology in naïve *Triaka* mice.

## *Esrp1*<sup>*Triaka*</sup> modulates intestinal immunopathology

Next, we addressed whether the *Esrp1*<sup>*Triaka*</sup>-dependent alterations observed at steady-state in *Triaka* mice may affect intestinal disease. Compared with WT controls, *Triaka* animals treated with dextran sodium sulfate (DSS) in drinking water for 7 days, a disease model of intestinal inflammation and damage, showed increased weight loss and pronounced shortening of the colon (*Figure 4A and B*). This was also reflected by higher disease scores, both clinically and histologically (*Figure 4—figure supplement 1A and B*). Notably, this phenotype was not due to an altered drinking behavior of *Triaka* animals (*Figure 4—figure supplement 1C*). In addition, *Triaka* mice were also more susceptible to chronic DSS-induced colitis (*Figure 4—figure supplement 1D*).

To assess a possible functional consequence of the differential expression of proliferation-associated genes in *Triaka* versus WT cIECs, we applied an in vivo wound-healing model and followed the repair of experimentally-induced intestinal lesions in the two groups of mice. In these settings, we

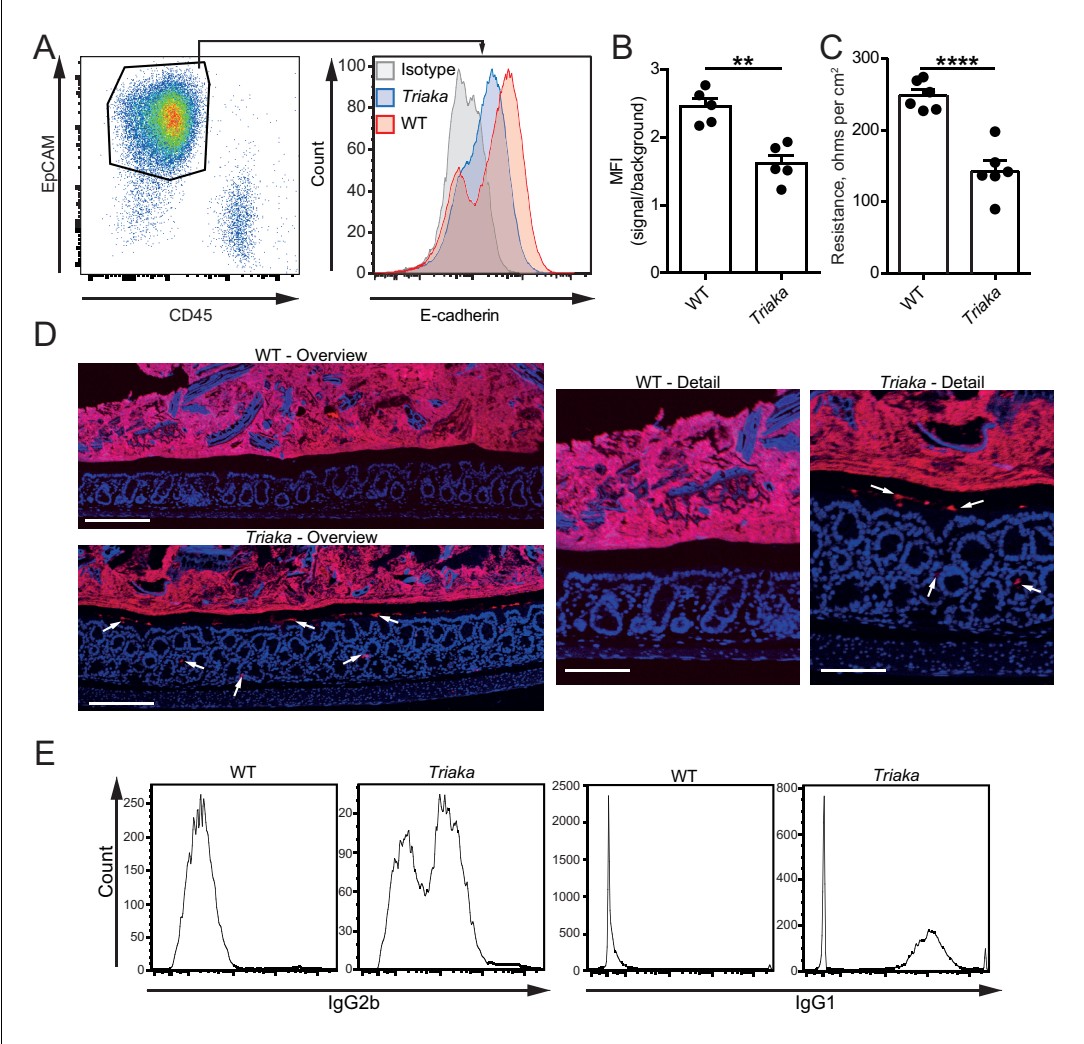

**Figure 3.** *Esrp1^Triaka* decreases the barrier function of the intestine. (**A**) Representative flow cytometry plot of colonic single cells stained for EpCAM and CD45 (left panel). Representative histogram indicating surface E-cadherin expression on EpCAM⁺ epithelial cells (right panel). (**B**) Median fluorescence intensity (MFI) of E-cadherin expression on EpCAM⁺ cells from WT and *Triaka* mice. Data from one representative experiment of three are shown (*n* = 5 mice per group). (**C**) Intestinal barrier resistance in colonic tissue of WT and *Triaka* was measured using an Ussing chamber (*n* = 6 mice per group, pooled from two independent experiments). (**D**) Immunofluorescence for 16S rRNA (in red) was performed on colon tissue. Nuclei were visualized with DAPI. White arrows indicate presence of bacterial products in mucus and crypts of *Triaka* mice. Representative pictures are shown (*n* = 7–8 mice per group). Scale bars: overview: 200 μm; detail: 100 μm. (**E**) Commensal-specific IgG1 and IgG2b antibodies were assessed in the serum of WT and *Triaka* animals. Representative flow cytometry histograms are shown (*n* = 10 per group). Data are shown as mean ± standard error of the mean. Statistics: (**B**) and (**C**) Student's *t* test. **p<0.01; ****p<0.0001.

DOI: https://doi.org/10.7554/eLife.28366.008

The following source data and figure supplements are available for figure 3:

**Source data 1.** Gene expression and pathway analysis.
DOI: https://doi.org/10.7554/eLife.28366.012

**Figure supplement 1.** *Esrp1^Triaka* alters the expression of genes associated with cell proliferation.
DOI: https://doi.org/10.7554/eLife.28366.009

**Figure supplement 2.** Normal colonic histomorphology in *Triaka* mice.
DOI: https://doi.org/10.7554/eLife.28366.010

**Figure supplement 3.** Normal small intestinal barrier integrity, fecal lipocalin-2 and albumin levels in *Triaka* mice.
DOI: https://doi.org/10.7554/eLife.28366.011

**Table 1.** Bacterial translocation and serum anti-commensal antibodies.

| | No. of WT mice | No. of *Triaka* mice | p-value |
|---|---|---|---|
| Penetration of bacteria in mucus or mucosa | | | |
| 16S rRNA | 0/9 | 6/8 | 0.0023 |
| Serum anti-commensal antibodies | | | |
| IgG1 | 1/10 | 7/10 | 0.0198 |
| IgG2b | 1/10 | 8/10 | 0.0055 |

Statistics: Fisher's exact test was performed. This table relates to **Figure 3**.
DOI: https://doi.org/10.7554/eLife.28366.013

observed diminished wound-healing in *Triaka* compared with WT mice (**Figure 4C**). In contrast to what we observed at steady-state, these data imply that *Esrp1*$^{Triaka}$ impairs the proliferative or regenerative capacity of cIECs during intestinal pathology.

Next, we tested whether the increased susceptibility to DSS-colitis and the lower repair ability of *Triaka* mice may influence cell proliferation and pathogenesis in a CRC model. We found that in azoxymethane (AOM) and DSS-treated *Triaka* mice, colorectal tumors were reduced in number and size compared with WT controls, thus providing additional evidence for a defective proliferation of *Triaka* IECs (**Figure 4D and E**). The tumor grade was not different between the two groups (**Figure 4F**). However, molecular analysis revealed more pronounced upregulation of matrix metallo-proteinase-3, granulocyte-colony stimulating factor, transforming growth factor β1 and interleukin-1α in *Triaka* versus WT tumors (**Figure 4—figure supplement 2**). These proteins are all established drivers of CRC progression, EMT and metastasis, thus indicating a more aggressive phenotype of *Triaka* tumors (**Li et al., 2016**; **Mroczko et al., 2006**; **Morris et al., 2014**; **Calon et al., 2012**; **Matsuo et al., 2009**). These features of *Esrp1*$^{Triaka}$ CRC lesions likely resulted from a partial EMT signature expressed by *Triaka* cIECs, prior to transformation (**Figure 4—figure supplement 3**).

To address the hypothesis that the above-observed phenotypes were indeed due to impaired cIECs proliferation in *Triaka* mice, we last analyzed intestinal Ki-67 expression in the colonic mucosa, after a 3 day treatment with DSS. Under these conditions of mild intestinal inflammation, which are not sufficient to induce epithelial erosions, *Triaka* cIECs showed reduced Ki-67 expression compared with WT cIECs (**Figure 4G and H**).

Taken together, these results indicate that altered *Esrp1*-regulated mRNA splicing deregulates cIEC function and affects the development of intestinal disease in *Triaka* mice, through a mechanism reducing IEC proliferation.

## *Esrp1*$^{Triaka}$ regulates the Wnt pathway via Gpr137 isoforms

We next investigated the molecular mechanisms downstream of *Esrp1*$^{Triaka}$ that lead to diminished proliferation of *Triaka* IECs. Among the 35 genes showing splicing isoforms with different relative frequency in *Triaka* versus WT cIECs, we chose genes for which only two isoforms were differently expressed. Of those, *Gpr137* emerged as a prominent candidate since it is involved in IEC proliferation (**Zhang et al., 2014**), although the mode of action of this orphan G protein-coupled receptor (GPCR) or of its isoforms is unknown. Thus, the two *Gpr137* splicing isoforms with different relative frequency in *Triaka* versus WT cIECs, *Gpr137_ENSMUST00000166115* and *Gpr137_EN-SMUST00000099776* (referred hereafter as *Gpr137_Long* and *Gpr137_Short*, respectively), were selected for further functional studies. By using our in vitro reporter system, we could validate that *Gpr137* is a splicing target of ESRP1 (**Figure 5—figure supplement 1**). Our RNA sequencing analysis indicated that *Gpr137_Long* is preferentially expressed in *Triaka* cIECs, whereas *Gpr137_Short* is predominant in WT cIECs. To address a potentially distinct role of *Gpr137_Long* versus *Gpr137_Short* in cIECs, CMT-93 cells were transduced with lentiviral vectors encoding these two *Gpr137* isoforms. We found a decreased cell proliferation and diminished epithelial monolayer tightness, as indicated by lower electrical resistance, in *Gpr137_Long*- versus *Gpr137_Short*-expressing cells. Importantly, in this particular assay the barrier function appeared to be distinct from the IEC proliferation. Indeed, although control vector-transduced cells proliferated less than cells transduced with vectors encoding *Gpr137* isoforms, they formed a tighter barrier (**Figure 5A and B**, **Figure 5—**

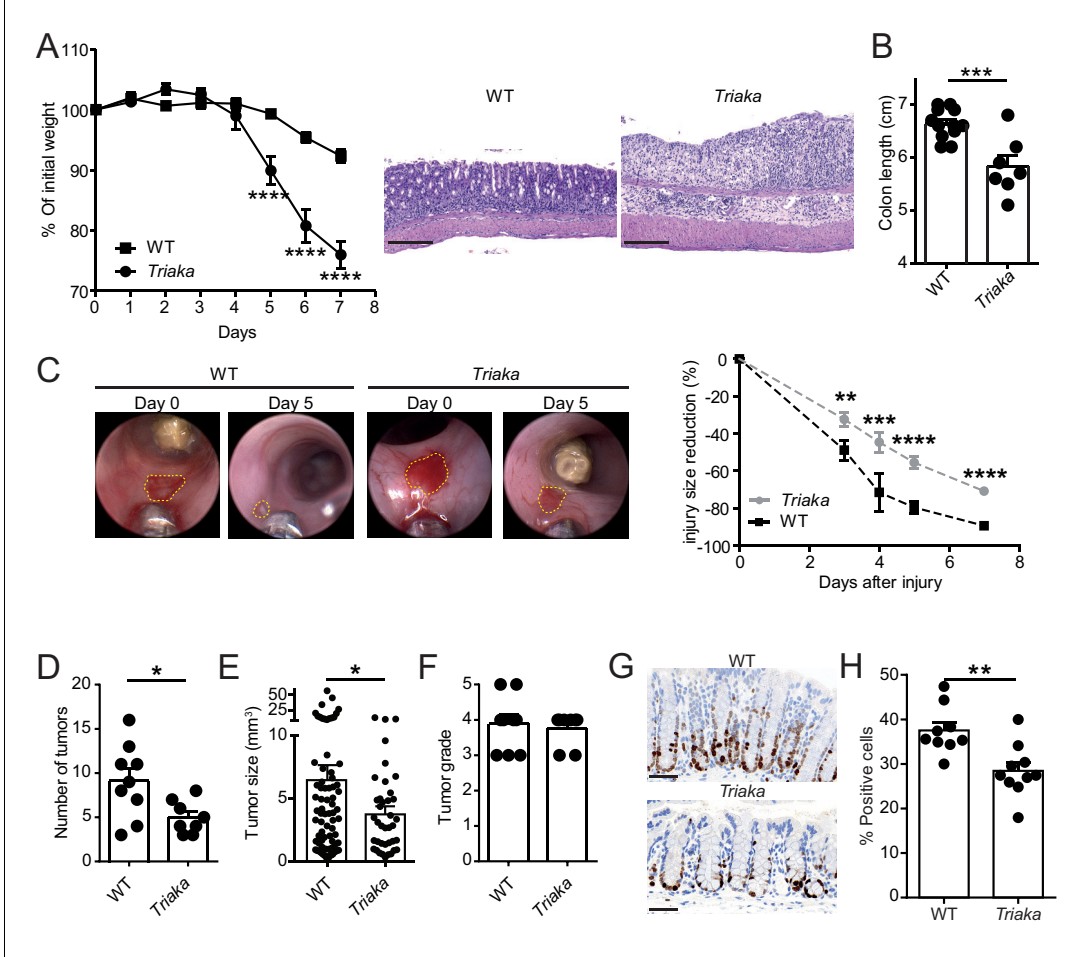

**Figure 4.** *Esrp1^Triaka* modulates the course of experimental intestinal immunopathology. (**A**) WT (*n* = 7) and *Triaka* (*n* = 4) mice were challenged with 2% DSS in the drinking water for 7 days. Weight loss was measured daily (left panel). One representative out of four different experiments is shown. Representative H&E slides from colonic sections illustrate the degree of pathology in the different groups 7 days after the start of DSS treatment (right panel). Scale bars: 200 μm. (**B**) Colon length was measured in DSS-treated mice (*n* = 7–12 mice per group). (**C**) A miniature forceps was used to induce injuries in the colonic mucosa of the indicated groups of mice. Wound-healing was monitored by colonoscopy. Representative pictures (left panel) and quantification of wound-healing over time (right panel) are shown. Right panel represents pooled data from four independent experiments (*n* = total of 17 wounds from 8 to 9 mice, per group). (**D**) WT and *Triaka* mice were treated with AOM/DSS and sacrificed. Number of tumors, (**E**) tumor size and (**F**) the highest tumor grade per mouse are shown, assessed 70 days after the initial AOM injection. For (**D**), (**E**) and (**F**), one representative experiment of two is shown (*n* = 8–9 mice per group). (**G**) WT and *Triaka* mice were treated with 2% DSS in the drinking water for 3 days and Ki-67 staining was performed. Representative pictures are shown and (**H**) Ki-67-positive cells were quantified (*n* = 9–10 mice per group). Statistics: (**A**) and (**C**) Two-way ANOVA with Bonferroni post-test, (**B**) and (**H**) Student's *t* test, (**D**) Mann-Whitney test and (**E**) Student's *t* test with Welch's correction. *p<0.05; **p<0.01; ***p<0.001; ****p<0.0001.

DOI: https://doi.org/10.7554/eLife.28366.014

The following figure supplements are available for figure 4:

**Figure supplement 1.** *Triaka* mice show increased susceptibility to experimental colitis.

DOI: https://doi.org/10.7554/eLife.28366.015

**Figure supplement 2.** More aggressive molecular signature in *Triaka* intestinal tumors.

DOI: https://doi.org/10.7554/eLife.28366.016

**Figure supplement 3.** Partial EMT signature in *Triaka* cIECs.

DOI: https://doi.org/10.7554/eLife.28366.017

*figure supplement 2A*). These effects of the two *Gpr137* isoforms on IEC proliferation and barrier integrity were independent of their relative expression levels and of possible cytotoxic effects (*Figure 5—figure supplement 2B and C*). Transduction with *Gpr137* isoforms also distinctively

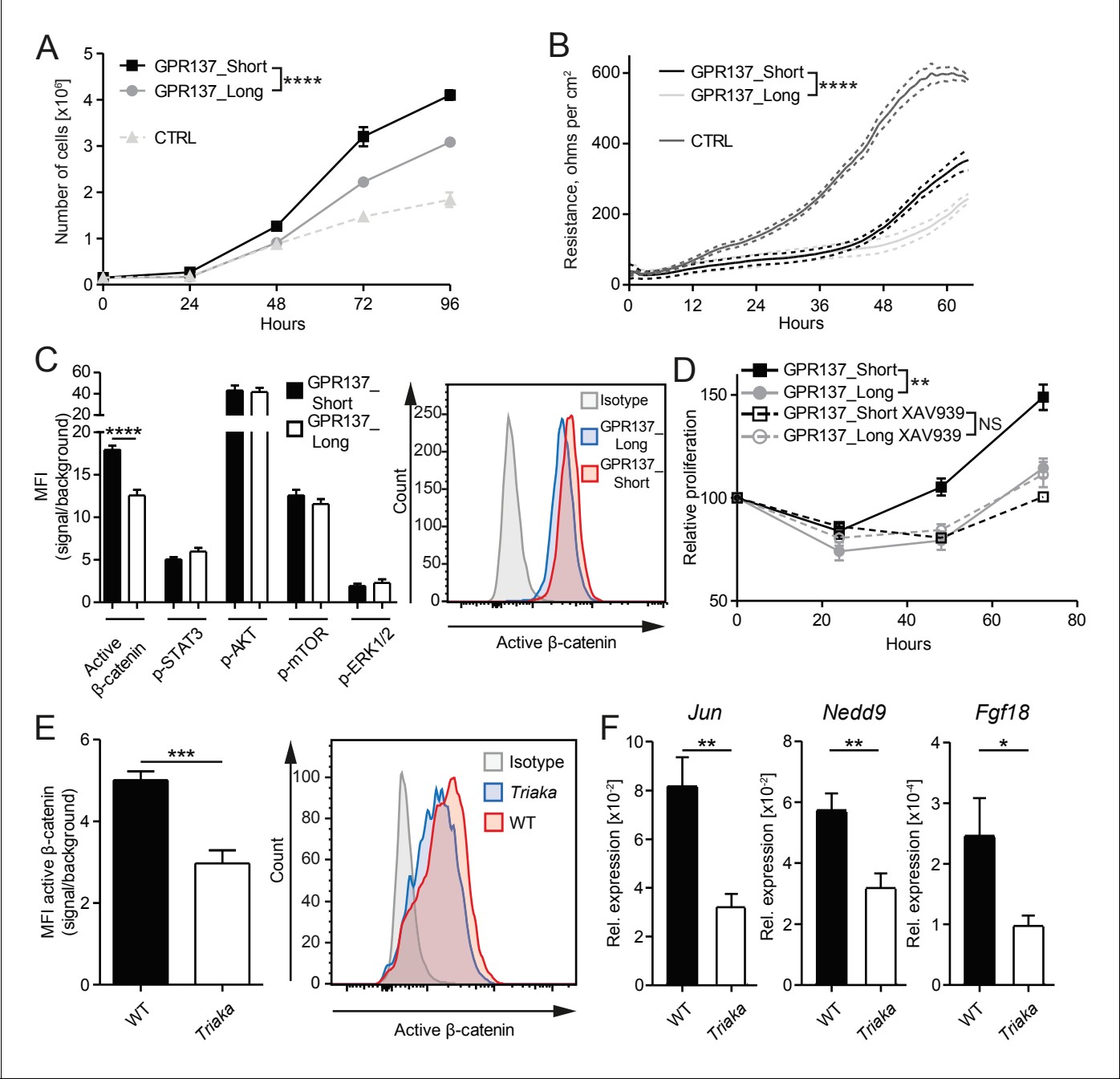

**Figure 5.** GPR137 isoforms differentially activate Wnt/β-catenin signaling to modulate epithelial cell function. (**A**) CMT-93 IECs were transduced with vectors encoding the indicated *Gpr137* isoforms or a control vector (CTRL) and live cells were counted daily by microscopy. (**B**) Alternatively, monolayer resistance was assessed using a cellZscope device. (**C**) Flow cytometry analysis was performed to assess the activity of selected signaling pathways in *Gpr137* isoform-transduced CMT-93 cells. Median fluorescence intensity (MFI) of the indicated proteins (left panel) and representative histogram indicating the level of active β-catenin (right panel) are shown. (**D**) A WST-1 assay was used to assess the relative proliferation of *Gpr137* isoform-transduced CMT-93 cells in the presence or absence of the Wnt/β-catenin signaling inhibitor XAV-939. Proliferation was normalized to control vector-transduced cells. (**E**) Expression levels of active β-catenin in primary cIECs of WT and *Triaka* mice (*n* = 7–8 mice per group) were measured by flow cytometry (left panel). A representative histogram is also shown (right panel). (**F**) Transcript levels of selected Wnt/β-catenin signaling target genes were measured in primary cIECs of WT and *Triaka* mice (*n* = 7–8 mice per group). Data shown: One representative experiment performed in (**A and D**) biological triplicates or (**B**) quadruplicates, and which was repeated four times (**A**) or twice (**B and D**). Pooled data from (**C**) three individual experiments performed in biological triplicates or (**E and F**) two individual experiments. Statistics: (**A**), (**B**) and (**D**) Two-way ANOVA with Bonferroni post-test. (**C**), (**E**) and (**F**) Student's *t* test. *p<0.05; **p<0.01; ***p<0.001; ****p<0.0001.

DOI: https://doi.org/10.7554/eLife.28366.018

*Figure 5 continued*

The following figure supplements are available for figure 5:

**Figure supplement 1.** Gpr137 is a splicing target of ESRP1.

DOI: https://doi.org/10.7554/eLife.28366.019

**Figure supplement 2.** *Gpr137* isoforms differently modulate the proliferation of epithelial cells.

DOI: https://doi.org/10.7554/eLife.28366.020

**Figure supplement 3.** Reduced Wnt/β-catenin signaling in *Esrp1^Triaka^*- compared with *Esrp1*^WT^-transduced cells.

DOI: https://doi.org/10.7554/eLife.28366.021

enhanced the proliferation of murine MC-38 and human Caco-2 IEC lines, thus validating our findings in CMT-93 cells (*Figure 5—figure supplement 2D and E*).

To dissect the differential effect of the two *Gpr137* isoforms on CMT-93 cells, we then investigated the activation of signaling pathways known to regulate epithelial cell proliferation or barrier function. Among the pathways examined, differences were only observed for Wnt/β-catenin signaling. Indeed, we found a one-third reduction in levels of active β-catenin protein in *Gpr137_Long*-versus *Gpr137_Short*-transduced cells (*Figure 5C*), which was validated in *Esrp1^Triaka^*- versus *Esrp1*^WT^-transduced cells (*Figure 5—figure supplement 3*). Furthermore, pharmacological inhibition of Wnt/β-catenin signaling abrogated the proliferative advantage of *Gpr137_Short*- over *Gpr137_Long*-transduced cells (*Figure 5D*). This suggested that *Gpr137* isoforms modify epithelial function via regulation of Wnt/β-catenin signaling. In line with these in vitro data, we measured ca. 40% reduced levels of active β-catenin, as well as diminished expression of Wnt target genes (*Herbst et al., 2014*) in cIECs of *Triaka* compared with WT mice (*Figure 5E and F*). Collectively, these results show that distinct *Gpr137* isoforms can differently modulate the Wnt/β-catenin pathway and IEC function, which may partially underlie the intestinal phenotype of *Triaka* mice.

## ESRP1 expression is reduced in the diseased human intestine

To address the general relevance of the findings from our in vitro and in vivo studies, we next assessed *ESRP1* expression in intestinal biopsies from Crohn's disease (CD) patients, after normalization to an epithelial-specific marker. Compared with non-inflamed paired biopsies, *ESRP1* levels were downregulated in the inflamed biopsies from CD intestines (*Figure 6A*). This correlated with reduced nuclear ESRP1 expression in the inflamed CD intestine, as measured by high-throughput automated immunohistochemistry quantification (*Figure 6B*).

We also performed a tissue microarray (TMA) based analysis of matched intestinal tissues from a cohort of 185 CRC patients, which revealed a gradual decrease in nuclear expression of ESRP1 protein during cancer progression (*Table 2*). Furthermore, low nuclear ESRP1 expression was associated with reduced patient survival (p=0.0456), larger tumors (p=0.0034), lymphatic invasion (p=0.0466), advanced pT-stage (p=0.02) and presence of nodal metastasis (p=0.016) (*Figure 6C* and *Figure 6— source data 1*). ESRP1 expression was determined to be an independent prognostic factor, after adjusting for the confounding effects of pT, pN, pM, tumor budding, and lymphatic invasion (*Figure 6—source data 2*). This indicates a possible tumor-suppressive role of ESRP1 in CRC. Finally, we analyzed in The Cancer Genome Atlas (TCGA) RNA sequencing data of intestinal tissue from CRC patients to evaluate the relative frequency of *GPR137* isoforms in the human intestine. We found a positive correlation between the expression of *ESRP1* and *GPR137_ENST00000539833* (referred hereafter as *hGPR137_Short*) in normal intestine tissue, which we further validated using independent samples (*Figure 6D* and *Figure 6—figure supplement 1A*). Moreover, there was an association between *hGPR137_Short* expression and transcript levels of Wnt target genes, suggesting a similar signaling downstream of GPR137 in humans and mice (*Figure 6—source data 3*).

In human CRC, the ratio between *hGPR137_Short* and *GPR137_ENST00000377702* (referred hereafter as *hGPR137_Long*) isoforms was altered compared with normal intestinal tissue (*Figure 6E*). Similarly, transcript levels of *Gpr137_Long* were increased in murine CRC tissue (*Figure 6—figure supplement 1B*). Importantly, a higher ratio of *hGPR137_Short* to *hGPR137_Long* transcripts predicted enhanced CRC patient survival, thereby indicating a protective role of the *ESRP1*-dependent *hGPR137_Short* isoform (*Figure 6F*). Indeed, higher expression levels of *hGPR137_Short* alone was also associated with better prognosis (*Figure 6—figure supplement 1C*).

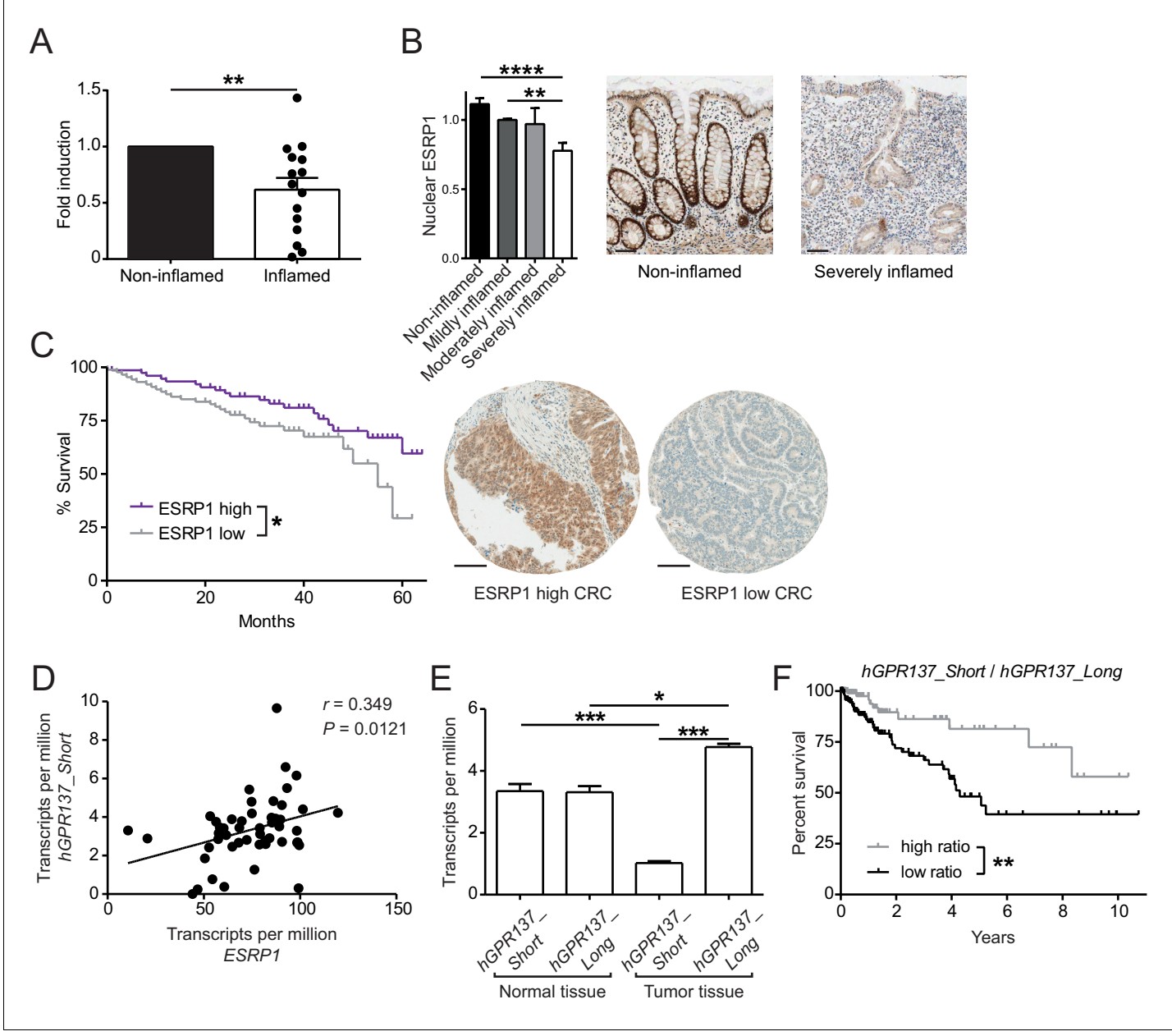

**Figure 6.** Expression of ESRP1 and *ESRP1*-dependent *hGPR137* isoforms is down-regulated in the diseased intestine and predicts CRC patient survival. (**A**) *ESRP1* transcript levels were measured in inflamed versus matched, non-inflamed intestinal biopsies from Crohn's disease (CD) patients and normalized to *EPCAM* expression. Normalized *ESRP1* transcript levels in the non-inflamed biopsy were set to one for each patient and fold induction was calculated for the corresponding inflamed biopsy (n = 15 samples per group). Data represent means ± standard error of the mean. (**B**) Immunohistochemistry was performed on intestinal tissue of CD patients to detect nuclear ESRP1, which was measured using automated quantification and normalized. Representative pictures are shown from a patient during remission and active disease, respectively (n = 31, 32, 7 and 14 biopsies per indicated group of cases). Data represent means ± standard error of the mean. Scale bars: 50 µm. (**C**) Kaplan-Meier survival curves of CRC patients with high (n = 77) or low (n = 88) expression of ESRP1 in tumor tissues. Representative IHC showing ESRP1-high and -low intestinal tumors. Scale bars: 100 µm. (**D**) Correlation between *ESRP1* and *hGPR137_Short* expression in normal tissue of the large intestine (n = 51). (**E**) *hGPR137_Short* and *hGPR137_Long* isoform expression in tumor (n = 647) versus normal (n = 51) tissue of the large intestine. Data represent means ± standard error of the mean. (**F**) Kaplan-Meier survival curves of CRC patients with a high (n = 142) or low (n = 261) ratio of *hGPR137_Short* to *hGPR137_Long* transcripts in tumor tissues. Statistics: (**A**) Wilcoxon signed-rank test, (**B**) One-way ANOVA with Bonferroni post-test, (**C**) and (**F**) Log-rank test, (**D**) Spearman correlation, (**E**) Kruskal-Wallis with Dunn's post-test. *p<0.05; **p<0.01; ***p<0.001; ****p<0.0001.

DOI: https://doi.org/10.7554/eLife.28366.022

The following source data and figure supplements are available for figure 6:

*Figure 6 continued on next page*

*Figure 6 continued*

**Source data 1.** Association of (nuclear) ESRP1 expression with clinicopathological features in 185 CRC patients.
DOI: https://doi.org/10.7554/eLife.28366.025
**Source data 2.** Univariate and multivariate survival analysis in 185 CRC patients.
DOI: https://doi.org/10.7554/eLife.28366.026
**Source data 3.** Correlation of *hGPR137_Short* with Wnt target genes.
DOI: https://doi.org/10.7554/eLife.28366.027
**Figure supplement 1.** Expression of *GPR137* isoforms in the healthy and diseased intestine.
DOI: https://doi.org/10.7554/eLife.28366.023
**Figure supplement 2.** ESRP1-dependent alternative mRNA splicing is required for epithelial integrity and intestinal homeostasis.
DOI: https://doi.org/10.7554/eLife.28366.024

These findings extend our in vivo data in *Esrp1^{Triaka}* mice and substantiate their relevance for the human intestine. They also suggest an involvement of *hGPR137* isoforms in human CRC tumorigenesis and progression.

Collectively, our data in mice and humans indicate that downregulation or loss of ESRP1 function results in dysregulation of alternative mRNA splicing in IECs. This leads to impaired epithelial cell integrity and contributes to intestinal pathology, possibly via altered *Gpr137/GPR137* isoform ratios (*Figure 6—figure supplement 2*).

## Discussion

Here, we describe a mouse model with hypomorphic ESRP1 function that shows impaired intestinal epithelial barrier integrity. This was associated with changes in the relative frequency of distinct mRNA isoforms in IECs and accompanied by an altered epithelial phenotype, as indicated by diminished surface E-cadherin expression. Of the 35 genes with splicing isoforms present at different ratios in *Triaka* versus WT primary cIECs, we identified two isoforms of *Gpr137* that differently affected IEC function via modulation of the Wnt signaling pathway. In support of these data from mouse models demonstrating a role for *Esrp1* in intestinal integrity, ESRP1 was downregulated in colonic tissue from CRC patients. Furthermore, the ratio of specific *GPR137* isoforms varied in tumor versus normal intestinal human samples, and expression of a specific *ESRP1*-dependent *GPR137* isoform predicted CRC patient survival. Little is known about GPR137, an orphan GPCR whose knockdown leads to reduced proliferation of several cancer cell lines, including colon cancer cells (*Zhang et al., 2014*). In particular, the function of distinct *Gpr137/GPR137* isoforms in IECs has not been addressed so far. GPR137 is ubiquitously and abundantly expressed in mouse tissues and is most closely related to OR51E2/PSGR, a prostate-specific GPCR (*Vanti et al., 2003*; *Regard et al., 2008*).

Interestingly, recent data suggest that GPR137 may be involved in EMT. Indeed, silencing of *GPR137* resulted in a downregulation of SNAI1/SNAIL and SNAI2/SLUG and a corresponding increase in E-cadherin expression in human prostate cancer cells (*Ren et al., 2016*). *Gpr137_Short* and *hGPR137_Short* are predicted to encode a protein with five transmembrane domains, whereas the products of *Gpr137_Long* and *hGPR137_Long* have a seven-transmembrane-spanning motif. Human and mouse GPR137 proteins have a 78% identity, yet the respective short and long isoforms

**Table 2.** Gradual loss of nuclear ESRP1 expression is associated with CRC progression

| Tissue type | Number of cases | ESRP1 expression (%) | | p-value |
| --- | --- | --- | --- | --- |
| | | Average | Median | |
| Normal | 26 | 75 | 75 | <0.0001 |
| Adenoma | 42 | 56.8 | 60 | |
| Carcinoma | 185 | 26.5 | 15 | |
| Lymph node metastasis | 68 | 9.1 | 0 | |

Statistics: Kruskal-Wallis test was performed. This table relates to **Figure 6**.
DOI: https://doi.org/10.7554/eLife.28366.028

we report here do not directly match between the two species. The molecular mechanisms by which distinct *Gpr137/GPR137* isoforms differently regulate Wnt/β-catenin signaling or possibly determine CRC survival have yet to be investigated.

Our data indicate a reduction of active β-catenin levels in primary *Esrp1^Triaka^* IECs and a subsequent downregulation of the transcription of some – but not all – Wnt target genes. Active β-catenin was similarly decreased in CMT-93 cells transduced with an *Esrp1^Triaka^*- or a *Gpr137_Long*-expressing lentiviral vector, compared to controls. The Wnt/β-catenin pathway is central to maintain tissue renewal and stem cell activity in IECs as well as epithelial cell proliferation in intestinal crypts. Genetic ablation of β-catenin in adult mice leads to intestinal epithelial stem cell (IESC) differentiation, thereby resulting in impaired intestinal homeostasis and fatal loss of intestinal function (*Fevr et al., 2007*). Reduced Wnt/β-catenin signaling likely underlies the diminished healing capacity of *Triaka* IECs and their reduced proliferation after transformation in the AOM/DSS model. In contrast, the Wnt/β-catenin pathway is aberrantly activated in most CRC, often caused by *APC* loss-of-function (*Clevers and Nusse, 2012*). Therefore, our study suggests a link between ESRP1 and Wnt/β-catenin signaling in the intestine.

Besides *Gpr137*, we also revealed other genes with altered splicing patterns in *Triaka* compared with WT cIECs, which likely also contribute to the intestinal phenotype in *Triaka* mice. Among those ESRP1 target genes, *Cd44* is probably the most studied for its intestinal function. CD44 and its isoforms negatively regulate IEC apoptosis (*Zeilstra et al., 2008*; *Lakshman et al., 2004*) and participate in various cellular functions, such as proliferation, adhesion, and migration (*Ponta et al., 2003*). Furthermore, *CD44v4* and *CD44v6* isoforms are overexpressed in neoplastic IECs of patients with familial adenomatous polyposis (*Zeilstra et al., 2014*). The expression of *Cd44v4* and *Cd44v6* isoforms in murine IESCs positively correlates with expression of LGR5, a marker of adult IESCs, and CD44 variant isoforms promote CRC in *Apc^Min/+^* mice (*Zeilstra et al., 2014*). Therefore, the decreased expression of CD44v4 protein in *Esrp1^Triaka^* IECs may likewise explain the reduced intestinal tumorigenesis observed in *Triaka* compared with WT mice.

Although our data indicate reduced splicing activity from the *Triaka* mutation, a possible neomorphic effect of ESRP1^Triaka^ cannot be fully excluded, and further investigation using other genetic models is required to examine this aspect.

ESRP1 has also been reported to restrict pluripotency in embryonic stem cells (*Fagoonee et al., 2013*), and it may as well regulate the differentiation of adult stem cells, including IESCs. Similarly, ESRP1 downregulation promotes cancer cell stemness and metastasis (*Preca et al., 2015*). Accordingly, frameshift mutations of *ESRP1* are found in 25% of microsatellite instability-positive CRC tumors (*Ivanov et al., 2007*). Furthermore, our results from a CRC cohort show that ESRP1 protein is gradually downregulated during the adenoma to carcinoma sequence in intestinal tumors and that few cells express it in lymph node metastases. These findings strengthen previous data suggesting a tumor-suppressive role of *ESRP1* in CRC cells (*Leontieva and Ionov, 2009*). Remarkably, ESRP1 appears to be involved early during CRC tumorigenesis, since it is already downregulated in adenoma. Reduction of ESRP1 function in adenomatous lesions may compromise the intestinal barrier, thereby facilitating the penetration of bacterial products and promoting inflammation-mediated CRC development (*Grivennikov et al., 2012*).

Our data in mice suggest that while AOM/DSS-induced *Triaka* tumors are smaller, they show a more aggressive molecular signature. We do not have a definitive explanation for these seemingly opposing results. However, ESRP1 is a key regulator of EMT – it maintains the epithelial phenotype of a cell – and ESRP1 downregulation leads to a shift towards a mesenchymal phenotype. Indeed, we find higher transcript levels of the EMT-promoting transcription factors *Zeb1* and *Zeb2* in *Triaka* compared with WT cIECs. As EMT is associated with reduced proliferation as well as increased motility and invasiveness of malignant cells, this may account for the phenotype of *Triaka* tumors (*Kalluri and Weinberg, 2009*; *Evdokimova et al., 2009*; *Hur et al., 2013*).

Although we investigated ESRP1 expression and not *ESRP1* mutations in human CRC, our in vitro data indicate a direct correlation between *ESRP1* expression and splicing activity mediated by ESRP1. Therefore, we believe that *Triaka* mice, characterized by decreased ESRP1 activity, represent a valid model to address the contribution of *Esrp1*-dependent mRNA splicing to intestinal function.

In summary, our results demonstrate a role for *Esrp1* in intestinal homeostasis and disease in mice. Our findings also suggest that loss of *ESRP1* may contribute to CRC and IBD development in humans. Moreover, we identified *GPR137* as a novel splicing target of ESRP1 and found that distinct

isoforms of GPR137 regulate intestinal homeostasis by modulating IEC function through the Wnt/β-catenin pathway. Additional studies investigating the role of *GPR137* and *ESRP1* in the intestinal epithelium are warranted to understand how distinct GPR137 isoforms impact on CRC patient survival and to reveal strategies to target the ESRP1-GPR137 axis.

## Materials and methods

### Patient selection

A multi-punch TMA of matched tumor, adenoma and normal tissue from 220 CRC well-characterized patients surgically treated from 2004 to 2007 at the Aretaieion University Hospital, University of Athens, Greece was used to evaluate changes in ESRP1 expression during tumor progression and impact on clinicopathological features (see also *Figure 6—source data 1*). Additionally, RNA*later* (Thermo Fisher Scientific, Waltham, MA) -stored and formalin-fixed paraffin-embedded biopsies from IBD patients were obtained from the Swiss IBD Cohort (http://www.ibdcohort.ch/) and from the Institute of Pathology of the University of Bern (Switzerland), respectively. Alternatively, CRC tissues were provided by the Tissue Bank Bern. The use of patient data and samples was approved by the Ethics Committee at the University of Athens, Greece and the Cantonal Ethics Committee of Bern.

### Evaluation of ESRP1 in CRC and IBD cases

Nuclear ESRP1 expression in CRC tissue was detected by immunohistochemistry (IHC) and scored by a pathologist (V.H.K.). In detail, each spot was evaluated for the percentage of epithelial cells showing ESRP1 expression (0–100%). Additionally, the intensity of staining was evaluated on a four point scale (0-3). Final scores were formed by multiplication of these two values (range 0–300) and evaluated in correlation with clinicopathological features and survival. For the Kaplan-Meier curves, Receiver Operating Characteristic curve analysis was performed to identify the optimal immunohistochemical cut-off score with lymph node metastasis as the endpoint. Based on this analysis, a cutoff value of 20% ESRP1 expression in tumors was chosen to separate ESRP1-high from -low tumors. Univariate and multivariate survival analyses were carried out using Cox proportional hazards regression. All significant variables in univariate analysis were entered into multivariate analysis. Hazard ratios and 95% CI were used to determine the effect size for the outcome overall survival.

Nuclear ESRP1 expression in intestinal epithelial cells (IECs) was quantified using Oncotopix Discovery (RRID:SCR_015690; Visiopharm, Hoersholm, Denmark) and applying a custom-made algorithm developed by the authors. In brief, inflammation grade was assessed for each biopsy on H&E-stained tissue sections by a board-certified pathologist (V.G.), in a single-blinded manner, and classified into no, mild, moderate and severe inflammation. Sequential sections were then stained for ESRP1 and analyzed.

The analysis algorithm was first designed to automatically distinguish IEC areas from stroma and background. Nuclear ESRP1 expression in IECs was then stratified into no, low, moderate and high expression and the area of these different intensity values was quantified for each biopsy. Thereafter, the area-corrected mean brown stain intensity value was calculated for each biopsy. Finally, intensity values were normalized to mildly inflamed biopsies on each slide to control for potential staining variability between slides.

### Immunohistochemistry and immunofluorescence of human and mouse tissue

IHC was performed on sections (2.5 µm) of IBD biopsies and on the TMA using the BondMax system (Leica, Wetzlar, Germany) in Bond Epitope Retrieval Solution 2 (citrate, pH 9.0) for 60 min at 95℃. Tissues were then stained for ESRP1 (1:50, Thermo Fisher Scientific, Waltham, MA). The Bond Polymer Refine Detection kit (Leica) was used for the detection of ESRP1-positive cells.

For IHC on formalin-fixed paraffin-embedded murine colon tissue, a heat-induced epitope retrieval was first performed for 18 min in a steam cooker in 1 mM Tris/1 mM EDTA (pH 9.0). A primary antibody detecting total CD44 (1:100, rat anti-mouse, Becton Dickinson, Franklin Lakes, NJ), CD44v4-containing isoforms (1:100, rat anti-mouse, eBioscience, Santa Clara, CA) or Ki-67 (1:200, Dako, Santa Clara, CA) were then used. Specific binding of primary antibodies was visualized using a

secondary antibody (1:200, goat anti-rat, Dako) followed by a tertiary antibody with linked horseradish peroxidase (HRP) as the enzyme (EnVision+, anti-goat, Dako), and 3,3'-diaminobenzidine as the chromogen. Staining intensity of positive cells was quantified using an automated scanner and software (Aperio, Sausalito, CA).

Bacterial 16S rRNA was detected with the eubacterial probe EUB338 (5'-GCTGCCTCCCGTAGGAGT-3') (*Lücker et al., 2007*) labeled with Alexa647 (Eurofins). Sections were counterstained with DAPI (1:2000, BioLegend, San Diego, CA) and placed in mounting medium (Dako). Fluorescence signal was detected using an Olympus IX81 confocal microscope combined with a FluoView FV1000 device (Olympus, Tokyo, Japan).

## Animals

All animal experiments were performed in accordance with institutional and federal regulations governing animal care and use and were approved by The Scripps Research Institute (TSRI) Institutional Animal Care and Use Committee (La Jolla, CA, USA) (IACUC protocols 07–0057 and 09–0079) and the Cantonal Veterinary Office of Bern (Switzerland) (protocols BE76-11 and BE130/14). Animal experiments were carried out in compliance with the ARRIVE reporting guidelines. All strains used were on a C57BL/6 background. C57BL/6J mice (RRID:IMSR_JAX:000664) were purchased from Jackson Laboratories and thereafter bred in-house. $Esrp1^{Triaka}$ mice (RRID:MGI:5515349) were generated at TSRI using *N*-ethyl-*N*-nitrosourea. For all experiments, non-randomized groups of 8–12 week old $Esrp1^{Triaka/Triaka}$ and C57BL/6J animals were either co-housed (for females) or soiled bedding was exchanged weekly (for males) 3–4 weeks prior to and during experiments.

## Identification of the $Esrp1^{Triaka}$ allele

The $Esrp1^{Triaka}$ allele was generated using *N*-ethyl-*N*-nitrosourea, as previously described (*Hoebe et al., 2003*). It was initially identified to have a defect in natural killer cell (NK) cytotoxicity as well as a hyperactivity and circling behavior. These phenotypes segregated in subsequent breeding and the name *Triaka* was retained for the neurobehavioral phenotype. Initial confinement of the mutation was made by outcrossing of $Esrp1^{Triaka/Triaka}$ mutation mice to C3H/HeN mice, followed by backcrossing of F1 hybrids to the $Esrp1^{Triaka/Triaka}$ stock. The mutation was mapped to proximal Chromosome 4, with a peak LOD of 2.71 at D4mit235. Fine mapping further defined a critical region from the centromere to 13.7 mega base pairs. Whole genome sequencing of a homozygous *Triaka* mouse using SOLiD$^{TM}$ (Thermo Fisher Scientific, Waltham, MA) revealed an A to G transition at position 828 of *Esrp1*, for which 86.3%, 74.9%, and 62.3% of coding/splicing sequence was covered at least $1\times$, $2\times$, or $3\times$, respectively. Validation sequencing of the critical region covered all nucleotides for which discrepancies were observed and confirmed the $Esrp1^{Triaka}$ mutation, which is located in exon 4 of 16 total exons of *Esrp1*. The circling behavior, which is incompletely penetrant among homozygous *Triaka* mice, was investigated in correlation with an auditory or neurodevelopmental defect. However, *Triaka* mice displayed normal hearing in tests for auditory brain stem response and distortion product otoacoustic emissions. In addition, there were no obvious abnormalities on histological sections of the neocortex, cerebellum, kidney, liver, lung, heart, pancreas, thymus and spleen. Adult (8 to 12 week old) $Esrp1^{Triaka}$ were slightly leaner than WT mice (19 ± 1 g versus 21 ± 1 g for females and 22 ± 2 g versus 26 ± 2 g for males, respectively), likely due to the hyperactive behavior. $Esrp1^{Triaka}$ mice were later found to be susceptible to acute challenge with 2% dextran sodium sulfate (DSS). Increased susceptibility to DSS was also observed in homozygous versus WT mice on a mixed C57BL/6J-C57BL/10J background. The *Triaka* strain is further described at the Southwestern Medical Center Mutagenetix database (http://mutagenetix.utsouthwestern.edu) (*Krebs et al., 2016*).

## Evaluation of colonic crypt depth and goblet cells numbers

Colonic crypt depth was assessed in longitudinally-sectioned crypts on H&E-stained sections. Slides were scanned and crypt depth was digitally measured in 20 crypts per mouse and averaged using the Pannoramic Viewer software (RRID:SCR_014424; 3DHISTECH Ltd., Budapest, Hungary). Colonic goblet cell numbers were assessed in longitudinally-sectioned crypts on PAS-stained sections using the Pannoramic Viewer software. The number of PAS-positive cells per crypt was averaged from 10 crypts per mouse.

## Fecal albumin and lipocalin-2 measurement

Fecal pellets were collected, weighed and resuspended in PBS containing 1% FBS. A sandwich ELISA for fecal albumin and lipocalin-2 was performed on 96-flat bottom Nunc MaxiSorp plates (eBioscience). A purified anti-mouse albumin capture antibody (Bethyl Laboratories, Montgomery, TX) and a secondary HRP-conjugated anti-mouse albumin antibody (Bethyl Laboratories) were used for the capture and detection of fecal albumin. A Substrate Reagent Pack (R&D Systems, Minneapolis, MN) was used for the color reaction and 1 M sulfuric acid was added to stop color development. Colorimetric signals were measured on a SpectraMax M2e device (Bucher Biotec AG, Basel, Switzerland). Fecal lipocalin-2 was measured using the mouse Lipocalin-2/NGAL MAb ELISA kit (R&D systems), according to the manufacturer's instructions.

## Isolation and RNA sequencing of colonic intestinal cells

Colonic tissues were harvested and thoroughly washed in ice-cold calcium- and magnesium-free Hank's Balanced Salt Solution (HBSS). Colons were cut into 0.5–1 cm small pieces and digested in pre-warmed calcium- and magnesium-free HBSS containing 25 mM HEPES (Sigma-Aldrich, St. Louis, MO), 2 mM DTT (Sigma-Aldrich) and 5 mM EDTA (Merck, Darmstadt, Germany) at 37°C for 20 min. Supernatants were filtered through 70 µm cell strainers (BD Biosciences, San Jose, CA). The purity of colonic IECs (cIECs) was assessed by flow cytometry and was generally ≥95% of cIECs (defined as EpCAM$^+$, CD45$^-$ cells) in $Esrp1^{Triaka}$ and WT mice. Frequencies of major immune cell populations among CD45$^+$ cells were analyzed in IEC preparations using the markers CD3, CD19 and CD11b and were comparable in both groups of mice. Isolated cIECs were pelleted and suspended in TRI-reagent (Sigma-Aldrich). For RNA sequencing, mRNA was purified and RNA concentration and integrity were assessed using a Bioanalyzer 2100 (Agilent, Santa Clara, CA) prior to cDNA synthesis and library preparation (TruSeq Stranded mRNA Sample Preparation, Illumina, San Diego, CA). The libraries were sequenced on an Illumina HiSeq2500 sequencer by the Next Generation Sequencing Platform of the University of Bern.

## Computational analysis of RNA sequencing data

Between 45.7 and 59.1 million read pairs were obtained per sample. The reads were mapped to the mouse reference genome (Ensembl m38, build 75) using TopHat v. 2.0.11 (RRID:SCR_013035) (*Kim et al., 2013*). We then used HTseq-count v. 0.6.1 (RRID:SCR_011867) (*Anders et al., 2015*) to count the number of reads per gene, and DESeq2 v. 1.4.5 (RRID:SCR_015687) (*Love et al., 2014*) to test for differential expression between groups of samples. To investigate alternative pre-mRNA splicing, we identified transcripts and quantified their expression levels as described in detail elsewhere (*Trapnell et al., 2012*). Specifically, TopHat v. 2.0.11 (*Kim et al., 2013*) was used to map the reads to the iGenomes reference (Ensembl NCBIM37 available from Illumina) for which an annotation file with Cufflinks-specific attributes is available. For each sample taken separately, transcripts were assembled with Cufflinks (RRID:SCR_014597) and then combined into a single assembly with Cuff-merge (RRID:SCR_015688), resulting in approximately 145,000 transcripts. We tested for differential expression between $Esrp1^{Triaka}$ and WT mice using Cuffdiff (RRID:SCR_001647) and visualized the results with CummeRbund (RRID:SCR_014568). In particular, we identified genes where the relative frequency of the isoforms from a given transcription start site (TSS) differed between $Esrp1^{Triaka}$ and WT mice (false discovery rate (FDR)-adjusted p-value<0.05), consistent with alternative pre-mRNA splicing.

## Pathway analysis

The outcome of the DESeq2 analysis was taken to perform gene set enrichment analysis (GSEA) using the SetRank method (RRID:SCR_015689) (*Simillion et al., 2017*). This algorithm first calculates the p-value of a gene set utilizing the ranking of its genes in the ordered list of p-value as determined by DESeq2. Next, it discards gene sets that have been initially flagged as significant, if their significance is merely due to the overlap with another gene set. Gene sets were derived from the following databases: REACTOME (RRID:SCR_003485) (*Croft et al., 2014*), Gene Ontology (RRID:SCR_006447) (*Ashburner et al., 2000*), LIPID MAPS (RRID:SCR_006579) (*Fahy et al., 2009*), PhosphoSite-Plus (RRID:SCR_001837) (*Hornbeck et al., 2012*), KEGG (RRID:SCR_012773) (*Kanehisa et al., 2014*), BIOCYC (RRID:SCR_002298) (*Karp et al., 2005*), ITFP (RRID:SCR_008119) (*Zheng et al., 2008*) and

WikiPathways (RRID:SCR_002134) (*Kelder et al., 2012*). Counts indicated in *Figure 3—source data 1* were normalized for differences in total sequencing depth using size factor normalization in DESeq2.

### Measurement of intestinal barrier resistance

Intestinal barrier integrity was measured ex vivo using an Ussing chamber (Dipl.-Ing. K. Mussler, Scientific Instruments, Aachen, Germany), as previously described (*Clarke, 2009*). In brief, colon tissue was isolated from *Triaka* and WT mice, washed in ice-cold calcium- and magnesium-free HBSS and then mounted in a 37°C warm, oxygen saturated and HBSS-containing Ussing chamber. Electrical resistance was measured after 10 to 15 min of equilibration time.

### Measurement of serum anti-commensal antibodies

To assess systemic antibodies against intestinal commensals, fecal pellets were collected and intestinal bacteria from pellets were cultured in brain-heart infusion medium for 24 hours. Autologous fecal bacterial antibody binding was measured by bacterial flow cytometry. Complement in the serum was heat-inactivated and serum was titrated on sodium azide-inactivated bacteria. Specific binding of IgG1 and IgG2b to bacteria was detected with fluorochrome-labeled antibodies against mouse IgG1 or IgG2b (BioLegend, San Diego, CA) on a FACSArray device (BD Biosciences).

### Induction and assessment of colitis in mice

For the induction of acute colitis, animals were given 2% DSS (MP Biomedicals, Santa Ana, CA) in the drinking water for 7 days, followed by regular water. For chronic colitis, DSS was given in 3 cycles with 5 days of DSS followed by 7 days of regular water. Animal weight was measured throughout the procedure. Intestinal inflammation was assessed in a single blinded manner on H&E-stained tissue sections by a board-certified pathologist (V.G.), as previously described (*Brasseit et al., 2016*). Clinical disease activity was quantified as follows: Weight loss, compared to initial weight (Score 0, <1% weight loss; score 1, 1–5% weight loss; score 2, 5–15% weight loss; score 4, 15–20% or more weight loss); Stool consistency (Score 0, normal stool; score 2, loose stool; score 4, diarrhea); Intestinal bleeding (Score 0, negative; score 2, positive).

### Colon wound-healing assay

Animals were anesthetized with isoflurane prior and during the experimental procedure. A miniature flexible biopsy forceps was used to inflict 2–3 mucosal injuries per mouse. The procedure was visualized using a miniature rigid endoscope system (Karl Storz, Tuttlingen, Germany). Wound-healing was continuously monitored by colonoscopy 3, 4, 5, and 7 days after the biopsy. To quantify the surface area of the excised mucosa and its subsequent regeneration, lesions were photographed and wound areas were measured using Photoshop (RRID:SCR_014199; Adobe Systems, San Jose, CA) and normalized to the forceps diameter.

### Induction and assessment of intestinal tumors in mice

The azoxymethane (AOM) (Sigma-Aldrich)/DSS model of CRC was used to induce tumors, as previously described (*De Robertis et al., 2011*; *Mertz et al., 2016*). In short, mice were injected intraperitoneally (i.p.) with AOM (10 mg/kg of body weight). After 7 days, mice were given 1% DSS in the drinking water for 5 days, followed by 7 days of regular water. Thereafter, animals received a second injection of AOM, which was followed by two further cycles of DSS and regular water. Mice were sacrificed 70 days after the first injection of AOM. Tumors were counted macroscopically and measured with a caliper by two independent observers. Tumor size was measured as previously described (*Neufert et al., 2007*). A board-certified pathologist evaluated the tumor grade in a single blinded manner. Grade was scored as follows: Grade 0: no tumor, healthy intestinal epithelium; Grade 1: adenoma with mild dysplasia; Grade 2: adenoma with moderate dysplasia; Grade 3: adenoma with severe dysplasia; Grade 4: adenoma with high grade dysplasia and infiltration of the lamina propria (carcinoma in situ); Grade 5: invasive carcinoma.

## Cytokine measurements

Tumors or adjacent tumor-free colonic tissue were suspended in RIPA buffer (Sigma-Aldrich, St. Louis, MO) and homogenized using a TissueLyzer II (Qiagen, Venlo, Netherlands). Samples were then centrifuged and the protein concentration in the supernatant was measured with a Bradford Protein Assay (Bio-Rad, Hercules, CA). Protein concentrations were adjusted to 1500 µg/ml for each sample. Cytokines were analyzed by Multiplexing LASER Bead Technology (Eve Technologies, Calgary, Canada).

## Flow cytometry

IECs or intraepithelial lymphocytes were filtered through a 70 µm cell strainer (BD Biosciences) to obtain a single-cell suspension. Live/dead cell discrimination was performed using DAPI (BioLegend). Intracellular staining was performed after fixation of cells with 4% paraformaldehyde for 5 min and permeabilization with a 90% methanol solution at 4°C for 30 min. Antibodies used for flow cytometry are listed in *Supplementary file 1*.

## Quantitative PCR analysis

RNA from isolated colonic IECs was extracted using TRI-reagent (Sigma-Aldrich). RNA was then reverse-transcribed into cDNA using an M-MLV Reverse Transcriptase (Promega, Fitchburg, WI). FastStart SYBR Green Master (Roche, Basel, Switzerland) and commercial primers specific for *Cd44t*, *Cd44v4/v5*, *Gapdh*, *Jun*, *Nedd9*, *Fgf18*, *ESRP1*, and *EPCAM* (Qiagen, Venlo, Netherlands), as well as self-designed primers for specific isoforms of *Uap1*, *Magi1*, and G*pr137* (*Supplementary file 2*) were used for the qPCR reaction. Reactions were performed and analyzed on a StepOnePlus Real-Time PCR System (Life Technologies, Carlsbad, CA). Unless indicated, transcript levels were normalized to *Gapdh* expression. For tissue biopsies from Crohn's disease patients, *ESRP1* transcript levels were normalized to *GAPDH* and *EPCAM* expression.

## Generation of exon trap vector and in vitro assessment of splicing activity

To generate a system to assess splicing events in vitro, an EcoRV restriction site was introduced into a commercially-available exon trap vector (MoBiTec, Göttingen, Germany) using the following primers: Fwd: 5'-GAGGCCCGATATCTTCAGACC-3'; Rev: 5'-GGTCTGAAGATATCGGGCCT-3'. A Firefly luciferase gene (Promega) lacking the ATG start codon was then cloned into this EcoRV site. The variable exon 5 of *Cd44* along with parts of the flanking introns were amplified using the following primers; Fwd: 5'-CGCGGGCTCGAGCATTGCAACAGATATAGAGACAGAATC-3'; Rev: 5'-CGCGGGGCGGCCGCCCTCTTTCAGGCTCTGCAGA-3'. Similarly, a region of *Fgfr2* containing exon IIIb and IIIc, the intron in-between, as well as parts of the flanking introns were amplified using the following primers; Fwd: 5'- CGCGGGCTCGAGGTCTGTTCTAGCACTACGGGGAT-3'; Rev: 5'-CGCGGGGCGGCCGCGCAGTATGTACCTGGCGAAC-3'. For the cloning of the region of *Gpr137* containing exons 2 and 3, we performed a first amplification using the following primers; Fwd: 5'-G TGATGGGGTATCTCTGCTCC-3'; Rev: 5'-CCTTGATGTAGCACCCTTGGG-3'. This amplicon was then used to perform a nested PCR reaction using the following primers; Fwd: 5'-CTCCCAGGTGG TGTTCAATAGGC-3'; Rev: 5'-CCAAGGTAGAGCCACC AACC-3'.

Amplicons were then cloned into the multiple cloning site of the modified exon trap vector. To generate vectors encoding *Triaka* or WT *Esrp1*, cDNA of *Esrp1*^WT^ and *Esrp1*^*Triaka*^ were amplified using the following primers: Fwd: 5'-CGCGGGGGGATCCGCCACCATGACGGCGTCTCCGGATTA-3'; Rev: 5'-CCCGCGGTCGACCTTAAATACAAACCCATTCTTTGGG-3'. Amplicons were then cloned into a pBRIT-HA/FLAG vector (Addgene, Cambridge, MA). A third vector containing a Renilla luciferase gene was also used to control for variations in the transfection efficiency (Promega). HEK-293 cells (RRID:CVCL_0045) were transiently transfected by calcium phosphate co-precipitation with the modified exon trap vector containing the cloned *Cd44* or *Fgfr2* regions, the *Esrp1*^WT^- or the *Esrp1*-^*Triaka*^-encoding vector, and the Renilla luciferase-encoding vector. Firefly and Renilla luciferase activities were measured 24 hours after transfection in an Infinite 200 PRO reader (Tecan, Männedorf, Switzerland), using the Dual-Luciferase Reporter Assay (Promega). All cell lines were routinely tested negative for mycoplasma contamination.

## Quantification of cell proliferation and epithelial monolayer resistance

CMT-93 cells (RRID:CVCL_1986) were cultured as recommended by ATCC. Cell proliferation was either assessed by manual cell counting or using a WST-1 assay (Roche). In brief, $8 \times 10^4$ cells were seeded into 6-well plates and cells were counted daily using a Neubauer chamber. Alternatively, 2– $8 \times 10^3$ cells were seeded into a 96-well plate and proliferation was measured using a WST-1 assay, according to the manufacturer's instructions.

To measure the trans-epithelial electrical resistance (TEER) of transduced CMT-93 cells, $4 \times 10^4$ cells were seeded onto rat tail collagen (Sigma Aldrich) coated 24-well plate ThinCerts™ inserts (Greiner Bio-One, Kremsmünster, Austria). These inserts were then placed into a cellZscope device (nanoAnalytics, Münster, Germany) and TEER was measured every hour for up to 72 hours.

## Construction and evaluation of tissue microarrays (TMAs)

Patient information on gender, age at diagnosis, tumor diameter, tumor location, post-operative therapy and disease-specific survival time was extracted from clinical records. The UICC TNM Classification 7th edition was used to assess the T- (pT), N- (pN), and M-stage (pM), lymphatic invasion (L), venous invasion (V), perineural invasion (Pn), tumor grade (G), histological subtype as well as tumor growth pattern. For each case, two TMA spots (diameter 0.6 mm) each of tumor center and tumor front and one spot of normal mucosa were stained for ESRP1 by immunohistochemistry (IHC). We excluded 35 cases based on insufficient material remaining on the tissue block. Final number of patients with invasive CRC was 185. From these patients, additional TMAs containing matched samples of normal colonic mucosa ($n = 26$), adjacent adenomas ($n = 42$) and lymph node metastases ($n = 68$) were constructed. Patient characteristics are indicated in *Figure 6—source data 1*.

## Lentiviral overexpression system

Lentiviruses encoding *Gpr137_Long* or *Gpr137_Short* were generated using HEK-293 cells as previously described (*Tschan et al., 2003*). In brief, HEK-293 cells were transfected with pMD2G, pMDLg/pRRE, pRSV-Rev and pLV (Cyagen Biosciences, Santa Clara, CA) encoding EGFP with either *Gpr137_Long*, *Gpr137_Short,* or a control sequence. HEK-293 cells were authenticated by short tandem repeat (STR) profiling and confirmed to be mycoplasma-negative monthly. Lentiviruses were then harvested to infect CMT-93 cells. Transduced CMT-93 cells were sort-purified based on EGFP expression and kept under selection pressure using 5 µg/ml Puromycin (Sigma-Aldrich). CMT-93 cells were originally purchased from the European Collection of Cell Cultures (ECACC no. 89111413) and confirmed to be mycoplasma-negative monthly.

A 4-hydroxytamoxifen inducible system previously described elsewhere (*Vince et al., 2007*; *Wang et al., 2006*) was modified to overexpress *Esrp1*$^{Triaka}$, *Esrp1*$^{WT}$ or EGFP. To this aim, we exchanged the *Hoxb8* cassette of the original vector with cDNA of *Esrp1*$^{Triaka}$ or *Esrp1*$^{WT}$. CMT-93 cells were then co-transduced with lentiviral vectors containing these different inserts and a lentiviral vector encoding a 4-hydroxytamoxifen responsive element. Co-transduced cells were kept under selective pressure using 5 µg/ml Puromycin and 200 µg/ml Hygromycin-B (Sigma-Aldrich). Expression of *Esrp1*$^{Triaka}$, *Esrp1*$^{WT}$ or EGFP was induced with 100 nM 4-hydroxytamoxifen.

## Data availability

RNA sequencing data form *Triaka* and WT IECs can be downloaded from the European Nucleotide Archive (ENA) (http://www.ebi.ac.uk/ena/data/search?query=PRJEB14221). All RNASeqV2 data for datasets COAD and READ (*Cancer Genome Atlas Network, 2012*) where downloaded though the TCGA data portal (COAD: tumor: $n = 480$; normal tissue: $n = 41$. READ: tumor: $n = 167$; normal tissue: $n = 10$) (https://tcga-data.nci.nih.gov/docs/publications/coadread_2012/; RRID:SCR_003193). From the RSEM results Tables, we extracted gene expression levels for *ESRP1*, for all eight reported *GPR137* isoforms, for Wnt target genes specific for CRC (*Herbst et al., 2014*), as well as survival data. All results are shown as transcripts per million obtained by multiplying the provided scaled estimates by $10^6$. The sequences of all *GPR137* isoforms were obtained from the FASTA file in the GAF bundle (hg19, June 2011).

For analysis of the survival data from the TCGA dataset, the X-tile software (RRID:SCR_005602) (*Camp et al., 2004*) was used to separate *GPR137_Short*-high from -low tumors. CRC Patients without survival data or with undetectable gene expression were excluded from the survival analysis.

## Statistical analysis

Sample size for in vivo studies was estimated by power analysis and adjusted for $\beta = 0.1$, with the assumption that differences between the groups were 1.5–2 fold. Statistical tests are two-sided and indicated in the figure legends. All statistical tests were performed with GraphPad Prism v.5.04 for Windows (RRID:SCR_002798; GraphPad Software, La Jolla, CA). Statistical tests were chosen based on the variation in each data group and on whether multiple comparisons were made. Groups with similar variance were compared using parametric tests, and groups with significantly different variations were analyzed using non-parametric tests. No mouse was excluded from the analysis. Only statistically significant differences are indicated in the figures. For all statistical analyses: $*p<0.05$; $**p<0.01$; $***p<0.001$; $****p<0.0001$.

## Acknowledgements

We thank the team of the Translational Research Unit of the Institute of Pathology, Micha Eichmann, Katharina Brandl, Silvia Rihs, Thomas Kaufmann, Alessandro Lugli, Stefan Wyder, Mario Tschan and Deborah Krauer for their excellent technical support and advice. Santos Franco and Ulrich Müller helped to assess auditory and neurodevelopmental defects in *Triaka* mice. Samples from IBD patients were obtained from the Swiss IBD Cohort (no. 3347CO-108792). We also like to extend our gratitude towards Mario Noti, Carsten Riether, Tilman Rau and Christian Schürch for critical comments.

## Additional information

### Competing interests

Andrew J Macpherson: Reviewing editor, *eLife*. The other authors declare that no competing interests exist.

### Funding

| Funder | Grant reference number | Author |
|---|---|---|
| Schweizerischer Nationalfonds zur Förderung der Wissenschaftlichen Forschung | 310030_138188 | Philippe Krebs |
| Universität Bern | Research Foundation | Philippe Krebs |
| National Institutes of Health | 5P01AI070167 | Bruce Beutler |
| Boehringer Ingelheim Fonds | | Lukas Franz Mager |
| Gertrud-Hagmann Foundation for Malignoma Research | | Lukas Franz Mager |
| National Institutes of Health | 5U19A100627 | Bruce Beutler |
| Schweizerischer Nationalfonds zur Förderung der Wissenschaftlichen Forschung | 314730_163086 | Philippe Krebs |

The funders had no role in study design, data collection and interpretation, or the decision to submit the work for publication.

### Author contributions

Lukas Franz Mager, Conceptualization, Resources, Formal analysis, Funding acquisition, Investigation, Visualization, Methodology, Writing—original draft, Writing—review and editing; Viktor Hendrik Koelzer, Formal analysis, Investigation, Methodology; Regula Stuber, Lester Thoo, Formal analysis, Investigation; Irene Keller, Data curation, Formal analysis, Investigation, Methodology; Ivonne Koeck, Investigation, Generation of ESRP1 inducible vectors; Maya Langenegger, Investigation, Generation of CD44t and CD44v histology data; Cedric Simillion, Software, Formal analysis; Simona P Pfister, Martin Faderl, Irina Tcymbarevich, Resources, Methodology; Vera Genitsch, Formal

analysis, Analysis of histology data (board-certified pathologist); Pascal Juillerat, Christoph Müller, Resources, Access to biobank samples (SIBDCS); Xiaohong Li, Investigation, Methodology; Yu Xia, Formal analysis, Identification of mutation, initial bioinformatics analysis; Eva Karamitopoulou, Resources, Generation of CRC tissue microarray with documentation of patients data; Ruth Lyck, Resources, Formal analysis, Investigation; Inti Zlobec, Resources, Formal analysis; Siegfried Hapfelmeier, Resources, Contribution to analysis of anti-commensal antibody; Rémy Bruggmann, Resources, Software; Kathy D McCoy, Andrew J Macpherson, Resources, Re-derivation of Triaka embryos for import; Bruce Beutler, Resources, Funding acquisition, Project administration; Philippe Krebs, Conceptualization, Resources, Data curation, Formal analysis, Supervision, Funding acquisition, Investigation, Methodology, Writing—original draft, Project administration, Writing—review and editing

## Author ORCIDs
Lukas Franz Mager  [iD]  http://orcid.org/0000-0002-7426-2842
Martin Faderl  [iD]  https://orcid.org/0000-0001-8807-6146
Rémy Bruggmann  [iD]  http://orcid.org/0000-0003-4733-7922
Christoph Müller  [iD]  https://orcid.org/0000-0002-3921-8678
Philippe Krebs  [iD]  http://orcid.org/0000-0003-4918-6654

## Ethics
Human subjects: The use of patient data and samples was approved by the Ethics Committee at the University of Athens, Greece and the Cantonal Ethics Committee of Bern.
Animal experimentation: All animal experiments were performed in accordance with institutional and federal regulations governing animal care and use and were approved by The Scripps Research Institute (TSRI) Institutional Animal Care and Use Committee (La Jolla, CA, USA) (IACUC protocols 07-0057 and 09-0079) and the Cantonal Veterinary Office of Bern (Switzerland) (protocols BE76-11 and BE130/14).

## Decision letter and Author response
Decision letter https://doi.org/10.7554/eLife.28366.044
Author response https://doi.org/10.7554/eLife.28366.045

---

# Additional files

## Supplementary files
• Supplementary file 1. Antibodies and conjugates used for flow cytometry analysis.
DOI: https://doi.org/10.7554/eLife.28366.029
• Supplementary file 2. Self-designed primers for qPCR analysis.
DOI: https://doi.org/10.7554/eLife.28366.030
• Transparent reporting form
DOI: https://doi.org/10.7554/eLife.28366.031

## Major datasets
The following dataset was generated:

| Author(s) | Year | Dataset title | Dataset URL | Database, license, and accessibility information |
|---|---|---|---|---|
| Mager L, Koelzer V, Stuber R, Thoo L, Keller I, Koeck I, Langenegger M, Simillion C, Pfister SP, Faderl M, Genitsch V, Tcymbarevich I, Juillerat P, Li X, Xia Y, Karamitopoulou E, Lyck R, Zlobec I, Hapfelmeier S, Bruggmann R, McCoy KD, Macpherson AJ, Müller C, Beutler B, Krebs P | 2017 | Assessment of transcript isoforms in Esrp1Triaka versus wild-type primary colonic epithelial cells | https://www.ebi.ac.uk/ena/data/view/PRJEB14221 | Publicly available at the EMDataBank (accession no. PRJEB14221) |

The following previously published datasets were used:

| Author(s) | Year | Dataset title | Dataset URL | Database, license, and accessibility information |
|---|---|---|---|---|
| The Cancer Genome Atlas Network | 2012 | TCGA-COAD | https://portal.gdc.cancer.gov/projects/TCGA-COAD | Publicly available from the NCI GDC Data Portal (https://portal.gdc.cancer.gov) |
| The Cancer Genome Atlas Network | 2012 | TCGA-READ | https://portal.gdc.cancer.gov/projects/TCGA-READ | Publicly available from the NCI GDC Data Portal (https://portal.gdc.cancer.gov) |

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
