## [Decision Letter]

Thank you for submitting your article "The ESRP1-GPR137 axis contributes to intestinal pathogenesis" for consideration by *eLife*. Your article has been favorably evaluated by Wendy Garrett (Senior Editor) and three reviewers, one of whom served as Guest Reviewing Editor. The reviewers have opted to remain anonymous.

The reviewers have discussed the reviews with one another and the Reviewing Editor has drafted this decision to help you prepare a revised submission. We hope you will be able to submit the revised version within two months.

Summary:

In order to examine the role of ESRP1 in the intestinal epithelium, the manuscript takes advantage of the *Esrp1^Triaka^* model that exhibits a point mutation in *Esrp1* resulting in a single amino acid substitution (M161V) since conventional KO mice as well as mice with a conditional deletion of ESRP1 in the skin epithelium are not viable. Differential splicing of *Gpr137* in *Triaka* mice is identified here as one of the key downstream events and is a mechanism of how *Esrp1* controls intestinal homeostasis. Interestingly, the Triaka mutation sensitizes mice to chemical colitis (DSS), presumably triggered by an increased intestinal permeability and decreased Wnt signaling. However, due to the defects in intestinal regeneration the incidence of colitis-associated tumors in the AOM/DSS model is reduced yet the tumors seem more aggressive. This is an interesting phenomenon because often-increased colitis causes more (not less) cancer. A correlation between *Esrp1* and *Gpr137* are confirmed in IBD and CRC patients providing evidence for *Esrp1* loss being correlated with poor prognosis in CRC.

Essential revisions:

1) A central point of the manuscript is the effect on intestinal barrier integrity upon decreased ESRP1 expression. Here the authors should extend their attempts to explain the underlying mechanism and should perform a targeted and more comprehensive analysis of tight junction proteins. Moreover, increased bacterial translocation and anti-commensal antibody response, but normal fecal albumin, lipocalin and FITC-dextran permeability may indicate specific defects in mucus.

Does *Esrp-1* regulate splicing in *Muc2* or any enzymes known for modifications/glycosylation of *Muc2*?

2) The authors suggest that the ESRP1-mutated tumors are more aggressive and show signs of EMT. This appears to only be supported by an expression analysis of E-cadherin. The existence of EMT features should be supported by the inclusion of additional markers.

3) The human data, which enriches the vast amount of granular experiments performed in order to study the *Esrp1^Triaka^* model, highlight that *Esrp1* deserves attention from the community and certainly warrants further studies. For the human data on survival tables, please highlight the patient characteristics and the exclusion of differences that might explain survival benefits.

Minor points:

1) While the authors clearly show altered expression of two known targets of Esrp1 and decide to study hypofunctional *Esrp1* in intestinal epithelial cells, the exact consequence of the Esrp1-M161V substitution and its effects on Esrp1 function remains difficult to determine. Specifically, while both *Cd44v5* and *Fgfr2-*IIIb expression serve as proof-of-principle and are diminished in *Esrp1^Triaka^* mice, the degree to which they are diminished differs vastly. Is it possible that reduced function of Esrp1 (albeit to different degrees for each individual target) or potentially additional 'off'-target effects (modifying some of the 35 identified Ersp1-targets) explain parts of the phenotype. Please expand discussion on these topics.

*Reviewer #1:*

The manuscript by Mager and colleagues addresses the function of ESRP1 in intestinal epithelial cells. Using various in vivo and ex vivo assays the authors demonstrate a role for ESRP1 in epithelial barrier integrity and intestinal cell proliferation leading to increased susceptibility to DSS induced colitis and reduction of intestinal tumors when ESRP1 is absent. The authors claim that differential expression of *Gp137* isoforms contributes to these phenotypes. This is an interesting study that contains a large amount of human data and several experimental mouse models based on the development of a novel knockout mouse. The study is overall well performed and the data is well presented. While the manuscript is a little premature in terms of definite mechanistic insight into the exact molecular mechanism, the authors provide a very interesting data that will be the basis for further experiments. The greatest concern I have, however, is that most of the data is very correlative.

The following points should be addressed to improve this study:

1) Is there any difference in the expression of components of the tight junctions? This would support the authors’ claim about a barrier defect.

2) Is there any clear evidence for the existence of EMT apart from changes in E-cadherin?

*Reviewer #2:*

Mager et al. address the role of *Esrp1* (splicing regulator) – *Gpr137* (orphan receptor) in regulation of intestinal homeostasis, colitis and cancer.

This study uses novel mouse line generated through ENU mutagenesis, a hypomorph allele of splicing regulator *Esrp1* – called *Triaka* mice. As opposed to complete whole body or cell type specific knockout, *Triaka* mice are relatively normal, which allows the subsequent analysis in health and disease.

Differential splicing of *Gpr137* in *Triaka* mice is identified here as one of the key downstream events and is a mechanism of how *Esrp1* controls intestinal homeostasis. *Triaka* mutation predisposes mice to chemical colitis (DSS), may be because of increased permeability and decreased Wnt signaling, but due to the defects in intestinal regeneration reduces AOM+DSS colitis associated cancer. This is an interesting phenomenon because often-increased colitis causes more (not less) cancer. IBD and cancer related-phenomenon, as well as correlation between *Esrp1* and *Gpr137* are confirmed in IBD and CRC patients. *Gpr137*, whose alternative splicing is regulated by *Esrp1*, is identified as a negative regulator of Wnt/b-catenin signaling, linking *Triaka* mutation with defective proliferation of epithelial cells. Overall, this manuscript is very solid on its novelty of the mouse models and characterization of molecular events associated with *Esrp1* regulated splicing and on phenotypical characterization of intestinal pathology in *Triaka* mice. It is a little bit thin on mechanistic details further connecting *Esrp1- Grp13*7- b-catenin to intestinal phenotypes, but I think the authors have provided a new mouse model, very well characterized in vivo phenotype, solid in vitro data and human data. It really looks like this manuscript can open up the field to more questions and studies, but this story presented here is already solid enough by itself. Minor criticism is listed below:

1) Increased bacterial translocation and anti-commensal antibody response, but normal fecal albumin, lipocalin and FITC-dextrane permeability may indicate specific defects in mucus. Does Esrp-1 regulate splicing in Muc2 or any enzymes known for modifications/glycosylation of Muc2?

*Reviewer #3:*

In the current manuscript Mager et al. study the role of ESRP1 in the intestinal epithelium. Since conventional KO mice as well as mice with a conditional deletion of ESRP1 in the skin epithelium proved to be neonatal lethal the authors utilize the *Esrp1^Triaka^* model that exhibits a point mutation in *Esrp1* resulting in a single amino acid substitution (M161V). In this model the authors find a reduced but not abolished alternative splicing activity for two known targets of *Esrp1 (Cd44v5* and *Fgfr2*-IIIb). The authors subsequently identify 35 target genes in *Esrp1^Triaka^* mice that show an altered frequency of splicing isoforms compared to Wt mice. The authors then show an impaired barrier function along with increased susceptibility to DSS-colitis in *Esrp1^Triaka^* mice and smaller/less frequent but more aggressive tumors in the AOM-DSS model. The authors then focus on altered expression of *Gpr137* isoforms resulting in differential Wnt pathway activation as a mechanistic basis. Moreover, the authors are able to support their studies with human data showing less *Esrp1* expression in inflamed IBD biopsies, as well as providing evidence for *Esrp1* loss being correlated with poor prognosis in CRC. The authors are further able to link in human CRC differential *Gpr137* isoform expression with altered survival.

The major limitation of this study lies in the utilization of the *Esrp1^Triaka^* model itself. While clearly the authors show altered expression of two known targets of *Esrp1* and conclude therefore to study hypofunctional *Esrp1* in intestinal epithelial cells, the exact consequence of the *Esrp1*-M161V substitution and therefore on *Esrp1* function remains difficult to determine. Therefore, the relevance of studying this specific point mutation in the absence of an association with human disease is at least somewhat limited. Specifically, while both *Cd44v5* and *Fgfr2*-IIIb expression that serve as proof of principle are diminished in *Esrp1^Triaka^* mice the degree to which they are diminished differs vastly. Therefore it cannot be fully addressed if indeed reduced function of *Esrp1* (albeit to different degrees for each individual target) or additional off-target effects (artificially modifying some of the 35 identified *Ersp1*-targets) explain parts of the phenotype. Have the authors considered at all to study *Esrp1* heterozygous KO mice (*Esrp1*+/-) or mice exhibiting a heterozygous conditional deletion within the intestinal epithelium? Alternatively, a conditional deletion of both alleles at a later time point e.g. 6 weeks of age utilizing e.g. Vcre-ERT2 mice might provide additional insight.

Despite this limitation one has to congratulate the authors for this current work and the insight it generates for a potential role of *Ersp1* in the intestinal epithelium and more specifically in CRC. The experiments are thoroughly performed and conclusions supported by multiple lines of evidence. The human data, additionally to the vast amount of granular experiments performed in order to study the *Esrp1^Triaka^* model, highlight that *Esrp1* deserves attention from the community and certainly warrants further studies. For the human data on survival tables highlighting the patient characteristics and the exclusion of differences that might per se explain survival benefits still need to be added. This reviewer enjoyed very much reading the manuscript.

---

## [Author Response]

Essential revisions:1) A central point of the manuscript is the effect on intestinal barrier integrity upon decreased ESRP1 expression. Here the authors should extend their attempts to explain the underlying mechanism and should perform a targeted and more comprehensive analysis of tight junction proteins. Moreover, increased bacterial translocation and anti-commensal antibody response, but normal fecal albumin, lipocalin and FITC-dextran permeability may indicate specific defects in mucus.

To better assess barrier integrity in *Triaka* mice, we consulted the literature to compile an extensive list of (intestinal) genes belonging to the occludine-, claudine-, tight junction protein (TJP)- and junctional adhesion molecule (JAM) family (1, 2). We then analyzed our RNA-sequencing data to assess the transcript levels of these 21 selected genes.

We found that E-cadherin (*Cdh1*) transcripts were most abundant, showing in wild-type (WT) colonic intestinal epithelial cells (cIECs) a ~2 fold higher expression level than messages from the next coming gene, *Cldn7*. Yet, *Cdh1* transcript levels were decreased in *Triaka* compared to WT cIECs (Author response image 1). These findings on different *Cdh1* expression could be independently validated by real time quantitative PCR analysis from additional *Triaka* and WT cIECs (Author response image 1). They also further corroborate our data indicating reduced E-cadherin protein expression on *Triaka* cIECs (Figure 3).

E-cadherin is an epithelial adherens junction protein with central function for the formation of adherens junctions and desmosomes. Consequently, constitutive *Cdh1*-deficiency in IECs is neonatal lethal (3), while E-cadherin loss in IECs of adult mice leads to intestinal disease due to loss of intestinal epithelial integrity (4). Furthermore, *Cdh1*-deficient IECs showed compromised barrier function (3). Taken together, it is thus conceivable that the reduction of E-cadherin expression on *Triaka* IECs accounts for the reduced intestinal tightness in *Esrp1*-mutant mice. As cIECs have been described to be more dependent on E-cadherin expression for their integrity than IECs from the small intestine (4), this may possibly explain why in *Triaka* mice the large intestine – but not the small intestine – shows reduced intestinal electrical resistance ex vivo (see main manuscript).

Relatively to E-cadherin, the other (tight) junction proteins were less expressed in mouse IECs. Surprisingly, *Cldn8* and *Tjp2* were upregulated in *Triaka* versus WT cIECs, with 1.23 fold and 1.96 fold increase, respectively (Author response image 1). This may represent compensatory changes subsequent to E-cadherin down-modulation, as previously reported for other cell-cell adhesion genes (5).

In summary, these results suggest that downregulation of E-cadherin is the most relevant alteration in junction protein expression in *Triaka* versus WT cIECs, which likely contributes to impaired barrier integrity in *Esrp1^Triaka^* mutant mice.

**Author response image 1. respfig1:** Expression analysis of selected genes with known function for intestinal barrier integrity. (**A**) RNA sequencing was performed on colonic intestinal epithelial cells (cIECs) from *Triaka* and WT mice to assess expression of the indicated genes (*n* = 4 mice per group). (**B**) Quantitative PCR was performed to determine *Cdh1* transcript levels in *Triaka* versus WT cIECs, after normalization to *Gapdh* expression (*n* = 13-14 mice per group). Statistics: (**A**) Benjamini-Hochberg adjusted p-values from test for differential gene expression. (**B**) Student’s *t* test. **, *P* < 0.01; ****, *P* < 0.0001.

Does Esrp-1 regulate splicing in Muc2 or any enzymes known for modifications/glycosylation of Muc2?

Muc2 and MUC2 modifying enzymes – ESRP1-dependent mRNA splicing:

To address this point, we have specifically compared the splicing pattern of Mucin 2 (*Muc2*) isoforms in *Triaka* versus WT cIECs. In agreement with the data represented in Figure 2, there was no difference in the relative expression of *Muc2* isoforms. This suggests that ESRP1 does not regulate the splicing of *Muc2* transcripts.

We next investigated whether ESRP1 may possibly affect the splicing of transcripts of genes involved in the post-translational modification of MUC2. MUC2 protein becomes heavily O-glycosylated (6, 7), a process that is controlled by different glycosyltransferases (8). The initiation of O-glycosylation is performed by peptidyl-GalNAc transferases which add GalNAc to cell surface and secreted proteins being processed in the Golgi apparatus (7).

Thus, we compiled a list of all peptidyl-GalNAc transferases (a total of 19 in mice) as well as other glycosyltransferases previously identified in mouse IECs (9, 10); this list included *Galnt1, Galnt2, Galnt3, Galnt4, Galnt5, Galnt6, Galnt7, Galnt9, Galnt10, Galnt11, Galnt12, Galnt13, Galnt14, Galnt15, Galnt16, Galnt17, Galnt18, Galnt19, Galnt20, St6galnac6, St6galnac2, St6gal1, St3gal6, St3gal4, Gal3st2, Fut2, Chst4, B4galnt2, B4galnt1, B3gnt7, B3gnt3, B3galt5, B4galt4, B4galt1, Gcnt3, C1galt1c1 and C1galt1*. Analysis of our RNA sequencing data for splicing events in transcripts of these genes – which was performed for those transcripts expressed in sufficient levels in murine cIECs for this kind of analysis – did not uncover potential targets of ESRP1-dependent splicing in our model. This is also indicated in Figure 2.

MUC2 function also depends on proteolytic cleavage i.e. for the expansion of the mucus layer at the border of the inner to outer mucus layer of the colon (11). MUC2 is cleaved either through endogenous (12, 13) or bacteria-derived enzymes (14). Since the precise identities of these endogenous enzymes are currently unknown (12), we assessed whether ESRP1-dependent splicing may target transcripts from genes encoding proteolytic enzymes present in the inner and other mucus layer (12). Of these analyzed genes (*Apeh, Blmh, Casp1, Cndp2, Dnpep, Ggh, Klk, Lap3, Mep1b, Npepps, Pa2g4, Park7, Pgcp, Prep, Psma2, Psma4, Psma5, Psma6, Psma7, Psmb1, Psmb2, Psmb3, Psmb4, Psmb5, Psmb6, Rnpep, Thop1, Usp5*), none were found to overlap with identified ESRP1 targets in our dataset (see also Figure 2).

Lastly, we also assessed datasets from previous studies analyzing ESRP1-dependent splicing events in skin, urethral or pituitary cells (15-17). Systematic search of all the above-mentioned genes did not reveal isoforms of *Muc2* or of genes encoding enzymes involved in MUC2 post-translational modifications that are spliced in an ESRP1-dependent manner. However, it has to be mentioned that some of the above-listed genes are preferentially expressed in the intestine and may therefore not have been detected in these different RNA sequencing studies.

In summary, these data suggest that *Muc2* or MUC2-modifying genes are not directly affected by ESRP1-mediated mRNA splicing.

Muc2 and MUC2 modifying enzymes – transcript levels:

We next surmised that *Esrp1^Triaka^* might have indirectly altered the relative expression of these different genes involved in post-translational modification of MUC2 (all listed above). We thus analyzed our RNA sequencing data to assess whether transcript levels of these genes were differently expressed in *Triaka* versus WT cIECs. Of the 65 analyzed genes, only few displayed significant changes, with minor changes in expression levels, i.e. ~30% up- or downregulation in *Triaka* versus WT cIECs (upregulated genes: *Galnt3*, 1.27 fold, *Lap3*, 1.3 fold change; downregulated genes: *St6galnac6*, 0.73 fold; *St3gal6*, 0.79 fold; *St3gal4*, 0.67 fold and *B3galt5*, 0.73 fold change; Author response image 2).

Finally, while gene ontology analysis disclosed that the expression of genes involved in cell cycle and proliferation was affected in *Triaka* compared to WT cIECs (Figure 3—figure supplement 1 and Figure 3—source data 1, main manuscript), no alterations were detected in pathways or gene sets involved in barrier function or mucus production/modification (Author response table 1 below).

As these changes in expression are rather marginal and since gene ontology analysis did not identify alterations in barrier function or mucus production/modification in *Triaka* cIECs, we conclude that the differences in transcript levels of *Galnt3, Lap3, St6galnac6, St3gal6, St3gal4, B3galt5* are likely not biologically relevant.

**Author response image 2. respfig2:** Expression analysis of selected genes with known function for post-translational modification of intestinal mucus. RNA sequencing was performed on colonic intestinal epithelial cells (cIECs) from *Triaka* and WT mice (*n* = 4 mice per group) to assess expression of the indicated genes involved in (**A**) glycosylation or (**B**) proteolytic cleavage of MUC2. Statistics: Benjamini-Hochberg adjusted p-values from test for differential gene expression.

**Author response table 1. resptable1:** Unaltered barrier function or mucus production/modification pathways in *Triaka* cIECs.

SetID	Database	Description	Change *Triaka* vs. WT
GO:0070254	GOBP	mucus secretion	Not significant
GO:0070255	GOBP	regulation of mucus secretion	Not significant
GO:0070256	GOBP	negative regulation of mucus secretion	Not significant
GO:0070257	GOBP	positive regulation of mucus secretion	Not significant
GO:0070701	GOCC	mucus layer	Not significant
GO:0070702	GOCC	inner mucus layer	Not significant
GO:0070703	GOCC	outer mucus layer	Not significant
GO:0006486	GOBP	protein glycosylation	Not significant
GO:0006487	GOBP	protein N-linked glycosylation	Not significant
GO:0006493	GOBP	protein O-linked glycosylation	Not significant
GO:0006517	GOBP	protein deglycosylation	Not significant
GO:0018242	GOBP	protein O-linked glycosylation via serine	Not significant
GO:0018243	GOBP	protein O-linked glycosylation via threonine	Not significant
GO:0018279	GOBP	protein N-linked glycosylation via asparagine	Not significant
GO:0033575	GOBP	protein glycosylation at cell surface	Not significant
GO:0033577	GOBP	protein glycosylation in endoplasmic reticulum	Not significant
GO:0033578	GOBP	protein glycosylation in Golgi	Not significant
GO:0060049	GOBP	regulation of protein glycosylation	Not significant
GO:0060050	GOBP	positive regulation of protein glycosylation	Not significant
GO:0060051	GOBP	negative regulation of protein glycosylation	Not significant
GO:0090283	GOBP	regulation of protein glycosylation in Golgi	Not significant
GO:0090284	GOBP	positive regulation of protein glycosylation in Golgi	Not significant
GO:0090285	GOBP	negative regulation of protein glycosylation in Golgi	Not significant
5894152	Reactome	O-glycosylation of TSR domain-containing proteins	Not significant
GO:0070830	GOBP	tight junction assembly	Not significant
GO:1902396	GOBP	protein localization to tight junction	Not significant
GO:2000810	GOBP	regulation of tight junction assembly	Not significant
GO:0005923	GOCC	tight junction	Not significant
GO:0007045	GOBP	cell-substrate adherens junction assembly	Not significant
GO:0034332	GOBP	adherens junction organization	Not significant
GO:0034333	GOBP	adherens junction assembly	Not significant
GO:0034334	GOBP	adherens junction maintenance	Not significant
GO:0071896	GOBP	protein localization to adherens junction	Not significant
GO:0005912	GOCC	adherens junction	Not significant
GO:0005913	GOCC	cell-cell adherens junction	Not significant
GO:0005914	GOCC	spot adherens junction	Not significant
GO:0005924	GOCC	cell-substrate adherens junction	Not significant

Muc2 and MUC2 modifying enzymes – protein levels in situ:

We also assessed the expression of MUC2 protein as a surrogate of the mucus layer in cIECs, at steady-state. The thickness of the inner MUC-2 layer was similar between *Triaka* and WT (Author response image 3). Taken together, these different analyses do not provide evidence for overtly altered or impaired MUC2 expression, modification or function in the *Triaka* versus WT intestine.

**Author response image 3. respfig3:** Analysis Mucin-2 (MUC2) layer. (**A**) Immunofluorescence for MUC2 (in green) was performed on colon tissue of the indicated strains. Nuclei were visualized with DAPI. Representative pictures are shown (scale bar: 100µm). (**B**) Thickness of the inner mucus layer was measured on MUC2 immunofluorescent stained slides (*n* = 4-5 WT and *Triaka* mice, with 3 individual measurements per mouse).

Current explanations for the reduced intestinal barrier in Triaka mice:

Our additional data presented above suggest that the diminished tightness of the *Triaka* intestine is caused by mechanisms different than a direct effect of *Esrp1^Triaka^* on the splicing pattern of tight junction or adherence junction genes, or of *Muc2* or MUC2-modifiers. In addition, the expression of the majority of these genes in IECs is not or only marginally affected by the *Triaka* mutation.

Therefore, the reduced intestinal barrier in *Triaka* mice is likely caused by a combination of the following three parameters:

i) Epithelial-to-mesenchymal transition (EMT) is a process by which epithelial cell-cell junctions are destabilized and the apical-basal polarity of epithelia is lost (18, 19). ESRP1 is a well-established regulator of EMT (20, 21), and our results provide several lines of evidence for EMT-associated phenotypes in *Triaka* IECs, including decreased proliferative capacity and decreased expression of E-cadherin (an adherens junction protein). Indeed, downregulation of E-cadherin expression helps the destabilization of cell adhesions and the loss of apical-basal polarity to increase cell motility during EMT (18, 22). E-cadherin downregulation is therefore considered a hallmark of EMT (18). Transcription factors regulating EMT that include SNAIL, SLUG, TWIST, and the ZEB family all repress E-cadherin expression during EMT (18). Of note, inhibition of ZEB1/2 promotes tight junction assembly of breast epithelial cells (23). In contrast to these EMT-promoting factors, *ESRP1* expression correlates with that of E-cadherin (24). *ESRP1* overexpression has been reported to prevent EMT, among others by maintaining high levels of E-cadherin and the localization of this protein at cell junctions, while *ESRP1* knockdown induced the opposite effects (25). Given the above-discussed role of E-cadherin for intestinal epithelial integrity, it is thus likely that EMT-driven downregulation of E-cadherin contributes to the diminished intestinal barrier of *Triaka* mice.

ii) Moreover, we reveal in our study that distinct ESRP1-specific GPR137 isoforms induced different trans-epithelial electrical resistance in monolayers of transduced intestinal epithelial CMT-93 cells. Indeed, CMT-93 cells transduced with *Gpr137_Long* – the isoform preferentially expressed in *Esrp1^Triaka^* IECs – displayed lower barrier integrity compared to cells transduced with the *Gpr137_Short* isoform. Thus, we conclude that alteration in the relative frequency of GPR137 isoform partly underlie the defective barrier in *Triaka* mice (Figure 5).

iii) In addition to this involvement of reduced E-cadherin expression and altered generation of GPR137 isoforms, there may be other mechanisms contributing to the diminished barrier integrity in *Triaka* mice. For instance, CD44 is also an important regulator of barrier function (26) and a promoter of EMT (25, 27). Moreover, distinct CD44 isoforms were shown to affect the barrier function in endothelial cells (28) and to induce E-cadherin loss at epithelial cell-cell junctions (25). This is already discussed in the manuscript:

“…Besides *Gpr137*, we also revealed other genes with altered splicing patterns in *Triaka* compared with WT cIECs, which likely also contribute to the intestinal phenotype in *Triaka* mice. Among those ESRP1 target genes, *Cd44* is probably the most studied for its intestinal function. CD44 and its isoforms negatively regulate IEC apoptosis (71; 34) and participate in various cellular functions, such as proliferation, adhesion, and migration (49).”

2) The authors suggest that the ESRP1-mutated tumors are more aggressive and show signs of EMT. This appears to only be supported by an expression analysis of E-cadherin. The existence of EMT features should be supported by the inclusion of additional markers.

To address this point, we compared the expression of additional epithelial and mesenchymal markers commonly used to assess EMT. Analysis of our RNA sequencing data indicated a consistent trend for downregulation or upregulation of several epithelial- or mesenchymal-regulated gene transcripts in *Triaka* compared to WT cIECs, respectively (Author response image 4).

These findings were further validated for several of these markers by qPCR.Expression of the EMT-inducing transcription factors *Zeb1* and *Zeb2* was increased in *Triaka* cIECs (Author response image 4). Correspondingly, *Cdh1* and *Esrp1* expression was downregulated in these cells, in agreement with previous studies (24, 25) (Author response image 1 and Author response image 4, respectively). In addition, we show in the manuscript that, compared to controls, *Triaka* intestinal tumors display increased expression of TGF-β1 protein, a well-known EMT inducer (32, 33) (Figure 4—figure supplement 2).

Taken together, these results indicate a more mesenchymal transcription pattern in *Triaka* cIECs, therefore suggesting that these cells underwent partial EMT. We added this information to the revised main manuscript (see also new figure: Figure 4—figure supplement 3):

“…These features of *Esrp1^Triaka^* CRC lesions likely resulted from a partial EMT signature expressed by *Triaka* cIECs, prior to transformation (Figure 4—figure supplement 3).”

**Author response image 4. respfig4:** Partial EMT signature in *Triaka* cIECs. (**A**) RNA sequencing was performed on colonic intestinal epithelial cells (cIECs) from *Triaka* and WT mice to assess expression of the indicated epithelial or mesenchymal marker genes (*n* = 4 mice per group). (**B**) Quantitative PCR was applied to measure transcript levels for *Zeb1, Zeb2* (mesenchymal genes) and *Esrp1* (epithelial gene) in WT and *Triaka* cIECs, after normalization to *Gapdh* expression (*n* = 9-14 mice per group). Statistics: (**B**) Student’s *t* test. *, P < 0.05, **, P < 0.01.

3) The human data, which enriches the vast amount of granular experiments performed in order to study the Esrp1^Triaka^ model, highlight that Esrp1 deserves attention from the community and certainly warrants further studies. For the human data on survival tables, please highlight the patient characteristics and the exclusion of differences that might explain survival benefits.

To specifically address these comments, we first performed a univariate analysis assessing ESRP1 expression or clinicopathological features relatively to patient survival. Parameters found to be statistically significant in this first univariate analysis were subsequently analyzed in a multivariate analysis. In this multivariate analysis, ESRP1 expression was found to be a significant predictor of CRC patient survival when controlling for other confounding factors. We accordingly updated this information in the revised manuscript as follows:

“Furthermore, low nuclear ESRP1 expression was associated with reduced patient survival (*P =* 0.0456), larger tumors (*P =* 0.0034), lymphatic invasion (*P =* 0.0466), advanced pT-stage (*P =* 0.02) and presence of nodal metastasis (*P =* 0.016) (Figure 6 and Figure 6—source data 1). ESRP1 expression was determined to be an independent prognostic factor, after adjusting for the confounding effects of pT, pN, pM, tumor budding, and lymphatic invasion (Figure 6—source data 2). This indicates a possible tumor-suppressive role of ESRP1 in CRC.”

In addition, we added a new table showing the results of the univariate and multivariate survival analysis (Figure 6—source data 2) and we updated the section on “Evaluation of ESRP1 in CRC and IBD cases” in the Materials and methods of the revised manuscript.

Minor points:1) While the authors clearly show altered expression of two known targets of Esrp1 and decide to study hypofunctional Esrp1 in intestinal epithelial cells, the exact consequence of the Esrp1-M161V substitution and its effects on Esrp1 function remains difficult to determine. Specifically, while both Cd44v5 and Fgfr2-IIIb expression serve as proof-of-principle and are diminished in Esrp1Triaka mice, the degree to which they are diminished differs vastly. Is it possible that reduced function of Esrp1 (albeit to different degrees for each individual target) or potentially additional 'off'-target effects (modifying some of the 35 identified Ersp1-targets) explain parts of the phenotype. Please expand discussion on these topics.

We presently do not have evidence that off-target effects may be caused by *Esrp1^Triaka^*. Nevertheless, it has to be mentioned here that as many as 14 of the 35 identified genes with different relative frequency of splicing isoforms in *Esrp1^Triaka^*compared to *Esrp1*^WT^ cIECs have been previously reported to be targets of *Esrp1* in tissues other than intestine (15-17, 34). This further supports the notion of a hypomorphic instead of a neomorphic activity (or gain-of-function) of ESRP1*^Triaka^*.

We currently do not have a definitive explanation as to why hypofunctional *Esrp1^Triaka^* has a distinct impact on *Cd44v5* and *Fgfr2*-IIIb splicing. These differences may be reflective of i) non-physiological in vitro conditions, e.g. due to the use of transfected cell lines, ii) target gene-specific effects of *Esrp1^Triaka^* (i.e. exons of transcripts from distinct genes may be spliced by ESRP1 at different rates. Indeed, *Esrp*-regulated splicing events have been reported to show variable sensitivity to the loss of *Esrp1* (and/or *Esrp2*) (15, 24)), iii) or they may also depend on the relative abundance of target pre-mRNA. The latter was observed, e.g., for the splicing of *Fgfr2*-IIIb that was not affected by the *Esrp1^Triaka^* mutation in vivo, at steady-state. Inflammatory conditions upregulate *Fgfr2* expression (35-37), and we accordingly observed an increase in *Fgfr2*-IIIb transcripts in cIECs from mice with mild colitis. Interestingly, these inflammatory conditions also revealed the splicing defects of ESRP1*^Triaka^*, as evidenced by reduced *Fgfr2*-IIIb levels in *Triaka* compared to WT cIECs (Author response image 5). Yet these results are preliminary, as it is not known how inflammation may further modulate *Esrp1* expression or function.

**Author response image 5. respfig5:** Availability of target transcripts may determine the effect of the *Esrp1^Triaka^* mutation. *Fgfr2*-IIIb expression levels were measured in EPCAM^+^-sorted colonic intestinal epithelial cells of the indicated strains and normalized to *Gapdh* expression. Intestinal epithelial cells were isolated from mice at steady-state or from animals that underwent a short 3 day-treatment with dextran sodium sulfate (indicated as “Inflammatory conditions”) (*n* = 9 mice per group for steady-state conditions and *n* = 7 mice per group during inflammation conditions. Statistics: (**B**) Mann-Whitney test. *, *P* < 0.05.

We also discuss now these points in the revised manuscript, respectively:

“…ESRP1*^Triaka^* however showed reduced levels of *Cd44v5* and *Fgfr2-*IIIb inclusion, with 1.8 and 3.3 fold induction, respectively (Figure 1). This variation in the extent of in vitro splicing of *Cd44* versus *Fgfr2* by ESRP1*^Triaka^* likely relates to the fact that *Esrp1*-regulated splicing events show distinct sensitivity to *Esrp1* loss (15, 24).”

“…Although our data indicate reduced splicing activity from the *Triaka* mutation, a possible neomorphic effect of ESRP1*^Triaka^* cannot be fully excluded, and further investigation using other genetic models is required to examine this aspect.”

Reviewer #1:[…] The following points should be addressed to improve this study:1) Is there any difference in the expression of components of the tight junctions? This would support the authors claim about a barrier defect.

Please refer to Essential Revision #1 as an answer to this comment.

2) Is there any clear evidence for the existence of EMT apart from changes in E-cadherin?

Please refer to Essential Revision #2 as an answer to this comment.

Reviewer #2:[…] Minor criticism is listed below:1) Increased bacterial translocation and anti-commensal antibody response, but normal fecal albumin, lipocalin and FITC-dextrane permeability may indicate specific defects in mucus. Does Esrp-1 regulate splicing in Muc2 or any enzymes known for modifications/glycosylation of Muc2?Please refer to Essential Revision #1 as an answer to this comment.Reviewer #3:[…] The major limitation of this study lies in the utilization of the Esrp1^Triaka^ model itself. While clearly the authors show altered expression of two known targets of Esrp1 and conclude therefore to study hypofunctional Esrp1 in intestinal epithelial cells, the exact consequence of the Esrp1-M161V substitution and therefore on Esrp1 function remains difficult to determine. Therefore, the relevance of studying this specific point mutation in the absence of an association with human disease is at least somewhat limited. Specifically, while both Cd44v5 and Fgfr2-IIIb expression that serve as proof of principle are diminished in Esrp1^Triaka^ mice the degree to which they are diminished differs vastly. Therefore it cannot be fully addressed if indeed reduced function of Esrp1 (albeit to different degrees for each individual target) or additional off-target effects (artificially modifying some of the 35 identified Ersp1-targets) explain parts of the phenotype. Have the authors considered at all to study Esrp1 heterozygous KO mice (Esrp1+/-) or mice exhibiting a heterozygous conditional deletion within the intestinal epithelium? Alternatively, a conditional deletion of both alleles at a later time point e.g. 6 weeks of age utilizing e.g. Vcre-ERT2 mice might provide additional insight.

Please refer to Minor Point #1 as a partial answer to this comment.

Heterozygous ESRP1 knockout (*Esrp1*^+/-^) mice have been reported not to show any phenotype or splicing defects. Specifically, epidermal splicing of *Arhgef11, Enah, Fgfr1-IIIb, or Fgfr3-IIIb* was altered in homozygous *Esrp1*^-/-^ compared to WT mice, yet it was not altered in heterozygous *Esrp1*^+/-^ animals (15). Accordingly, it is unlikely that *Esrp1*^+/-^ mice show an intestinal phenotype.

References

1) Ulluwishewa D, Anderson RC, McNabb WC, Moughan PJ, Wells JM, Roy NC. Regulation of tight junction permeability by intestinal bacteria and dietary components. J Nutr. 2011;141(5):769-76. doi: 10.3945/jn.110.135657.

2) Luissint AC, Nusrat A, Parkos CA. JAM-related proteins in mucosal homeostasis and inflammation. Semin Immunopathol. 2014;36(2):211-26. doi: 10.1007/s00281-014-0421-0.

3) Bondow BJ, Faber ML, Wojta KJ, Walker EM, Battle MA. E-cadherin is required for intestinal morphogenesis in the mouse. Developmental biology. 2012;371(1):1-12. doi: 10.1016/j.ydbio.2012.06.005.

4) Schneider MR, Dahlhoff M, Horst D, Hirschi B, Trulzsch K, Muller-Hocker J, Vogelmann R, Allgauer M, Gerhard M, Steininger S, Wolf E, Kolligs FT. A key role for E-cadherin in intestinal homeostasis and Paneth cell maturation. PloS one. 2010;5(12):e14325. doi: 10.1371/journal.pone.0014325.

5) Chen A, Beetham H, Black MA, Priya R, Telford BJ, Guest J, Wiggins GA, Godwin TD, Yap AS, Guilford PJ. E-cadherin loss alters cytoskeletal organization and adhesion in non-malignant breast cells but is insufficient to induce an epithelial-mesenchymal transition. BMC cancer. 2014;14:552. doi: 10.1186/1471-2407-14-552.

6) Johansson ME, Ambort D, Pelaseyed T, Schutte A, Gustafsson JK, Ermund A, Subramani DB, Holmen-Larsson JM, Thomsson KA, Bergstrom JH, van der Post S, Rodriguez-Pineiro AM, Sjovall H, Backstrom M, Hansson GC. Composition and functional role of the mucus layers in the intestine. Cellular and molecular life sciences: CMLS. 2011;68(22):3635-41. doi: 10.1007/s00018-011-0822-3.

7) Arike L, Hansson GC. The Densely O-Glycosylated MUC2 Mucin Protects the Intestine and Provides Food for the Commensal Bacteria. Journal of molecular biology. 2016;428(16):3221-9. doi: 10.1016/j.jmb.2016.02.010.

8) Bennett EP, Mandel U, Clausen H, Gerken TA, Fritz TA, Tabak LA. Control of mucin-type O-glycosylation: a classification of the polypeptide GalNAc-transferase gene family. Glycobiology. 2012;22(6):736-56. doi: 10.1093/glycob/cwr182.

9) Johansson ME, Jakobsson HE, Holmen-Larsson J, Schutte A, Ermund A, Rodriguez-Pineiro AM, Arike L, Wising C, Svensson F, Backhed F, Hansson GC. Normalization of Host Intestinal Mucus Layers Requires Long-Term Microbial Colonization. Cell host & microbe. 2015;18(5):582-92. doi: 10.1016/j.chom.2015.10.007.

10) Zhang L, Tian E, Ten Hagen KG. UDP-N-Acetyl-Α-D-Galactosamine: Polypeptide N-Acetylgalactosaminyltransferases (ppGalNAc-Ts). In: Taniguchi N, Honke K, Fukuda M, Narimatsu H, Yamaguchi Y, Angata T, editors. Handbook of Glycosyltransferases and Related Genes. Tokyo: Springer Japan; 2014. p. 495-511.

11) Johansson ME, Larsson JM, Hansson GC. The two mucus layers of colon are organized by the MUC2 mucin, whereas the outer layer is a legislator of host-microbial interactions. Proc Natl Acad Sci U S A. 2011;108 Suppl 1:4659-65. doi: 10.1073/pnas.1006451107.

12) Johansson ME, Phillipson M, Petersson J, Velcich A, Holm L, Hansson GC. The inner of the two Muc2 mucin-dependent mucus layers in colon is devoid of bacteria. Proc Natl Acad Sci U S A. 2008;105(39):15064-9. doi: 10.1073/pnas.0803124105.

13) Schutte A, Ermund A, Becker-Pauly C, Johansson ME, Rodriguez-Pineiro AM, Backhed F, Muller S, Lottaz D, Bond JS, Hansson GC. Microbial-induced meprin β cleavage in MUC2 mucin and a functional CFTR channel are required to release anchored small intestinal mucus. Proc Natl Acad Sci U S A. 2014;111(34):12396-401. doi: 10.1073/pnas.1407597111.

14) Lidell ME, Moncada DM, Chadee K, Hansson GC. Entamoeba histolytica cysteine proteases cleave the MUC2 mucin in its C-terminal domain and dissolve the protective colonic mucus gel. Proc Natl Acad Sci U S A. 2006;103(24):9298-303. doi: 10.1073/pnas.0600623103.

15) Bebee TW, Park JW, Sheridan KI, Warzecha CC, Cieply BW, Rohacek AM, Xing Y, Carstens RP. The splicing regulators Esrp1 and Esrp2 direct an epithelial splicing program essential for mammalian development. *eLife*. 2015;4. doi: 10.7554/*eLife*.08954.

16) Bebee TW, Sims-Lucas S, Park JW, Bushnell D, Cieply B, Xing Y, Bates CM, Carstens RP. Ablation of the epithelial-specific splicing factor Esrp1 results in ureteric branching defects and reduced nephron number. Developmental dynamics: an official publication of the American Association of Anatomists. 2016;245(10):991-1000. doi: 10.1002/dvdy.24431.

17) Lekva T, Berg JP, Lyle R, Heck A, Ringstad G, Olstad OK, Michelsen AE, Casar-Borota O, Bollerslev J, Ueland T. Epithelial splicing regulator protein 1 and alternative splicing in somatotroph adenomas. Endocrinology. 2013;154(9):3331-43. doi: 10.1210/en.2013-1051

en.2013-1051 [pii].

18) Lamouille S, Xu J, Derynck R. Molecular mechanisms of epithelial-mesenchymal transition. Nat Rev Mol Cell Biol. 2014;15(3):178-96. doi: 10.1038/nrm3758.

19) Huang RY, Guilford P, Thiery JP. Early events in cell adhesion and polarity during epithelial-mesenchymal transition. Journal of cell science. 2012;125(Pt 19):4417-22. doi: 10.1242/jcs.099697.

20) Warzecha CC, Carstens RP. Complex changes in alternative pre-mRNA splicing play a central role in the epithelial-to-mesenchymal transition (EMT). Semin Cancer Biol. 2012;22(5-6):417-27. doi: 10.1016/j.semcancer.2012.04.003

S1044-579X(12)00059-4 [pii].

21) Yang Y, Park JW, Bebee TW, Warzecha CC, Guo Y, Shang X, Xing Y, Carstens RP. Determination of a Comprehensive Alternative Splicing Regulatory Network and Combinatorial Regulation by Key Factors during the Epithelial-to-Mesenchymal Transition. Molecular and cellular biology. 2016;36(11):1704-19. doi: 10.1128/MCB.00019-16.

22) Kalluri R, Weinberg RA. The basics of epithelial-mesenchymal transition. J Clin Invest. 2009;119(6):1420-8. doi: 10.1172/JCI39104.

23) Elsum IA, Martin C, Humbyert PO. Scribble regulates an EMT polarity pathway through modulation of MAPK-ERK signaling to mediate junction formation. Journal of cell science. 2013;126(Pt 17):3990-9. doi: 10.1242/jcs.129387.

24) Warzecha CC, Sato TK, Nabet B, Hogenesch JB, Carstens RP. ESRP1 and ESRP2 are epithelial cell-type-specific regulators of *FGFR2* splicing. Molecular cell. 2009;33(5):591-601. doi: S1097-2765(09)00069-0 [pii]

10.1016/j.molcel.2009.01.025.

25) Brown RL, Reinke LM, Damerow MS, Perez D, Chodosh LA, Yang J, Cheng C. CD44 splice isoform switching in human and mouse epithelium is essential for epithelial-mesenchymal transition and breast cancer progression. J Clin Invest. 2011;121(3):1064-74. doi: 10.1172/JCI44540.

26) Kirschner N, Haftek M, Niessen CM, Behne MJ, Furuse M, Moll I, Brandner JM. CD44 regulates tight-junction assembly and barrier function. The Journal of investigative dermatology. 2011;131(4):932-43. doi: 10.1038/jid.2010.390.

27) Xu H, Tian Y, Yuan X, Wu H, Liu Q, Pestell RG, Wu K. The role of CD44 in epithelial-mesenchymal transition and cancer development. OncoTargets and therapy. 2015;8:3783-92. doi: 10.2147/OTT.S95470.

28) Zhang P, Fu C, Bai H, Song E, Dong C, Song Y. CD44 variant, but not standard CD44 isoforms, mediate disassembly of endothelial VE-cadherin junction on metastatic melanoma cells. FEBS Lett. 2014;588(24):4573-82. doi: 10.1016/j.febslet.2014.10.027.

29) Zeilstra J, Joosten SP, Dokter M, Verwiel E, Spaargaren M, Pals ST. Deletion of the WNT target and cancer stem cell marker CD44 in Apc(Min/+) mice attenuates intestinal tumorigenesis. Cancer research. 2008;68(10):3655-61. doi: 10.1158/0008-5472.CAN-07-2940.

30) Lakshman M, Subramaniam V, Jothy S. CD44 negatively regulates apoptosis in murine colonic epithelium via the mitochondrial pathway. Experimental and molecular pathology. 2004;76(3):196-204. doi: 10.1016/j.yexmp.2003.12.009.

31) Ponta H, Sherman L, Herrlich PA. CD44: from adhesion molecules to signalling regulators. Nat Rev Mol Cell Biol. 2003;4(1):33-45. doi: 10.1038/nrm1004.

32) Miettinen PJ, Ebner R, Lopez AR, Derynck R. TGF-β induced transdifferentiation of mammary epithelial cells to mesenchymal cells: involvement of type I receptors. The Journal of cell biology. 1994;127(6 Pt 2):2021-36.

33) Xu J, Lamouille S, Derynck R. TGF-β-induced epithelial to mesenchymal transition. Cell research. 2009;19(2):156-72. doi: 10.1038/cr.2009.5.

34) Dittmar KA, Jiang P, Park JW, Amirikian K, Wan J, Shen S, Xing Y, Carstens RP. Genome-wide determination of a broad ESRP-regulated posttranscriptional network by high-throughput sequencing. Molecular and cellular biology. 2012;32(8):1468-82. doi: 10.1128/MCB.06536-11.

35) Huang JJ, Joh JW, Fuentebella J, Patel A, Nguyen T, Seki S, Hoyte L, Reshamwala N, Nguyen C, Quiros A, Bass D, Sibley E, Berquist W, Cox K, Kerner J, Nadeau KC. Eotaxin and FGF enhance signaling through an extracellular signal-related kinase (ERK)-dependent pathway in the pathogenesis of Eosinophilic esophagitis. Allergy Asthma Clin Immunol. 2010;6(1):25. doi: 10.1186/1710-1492-6-25.

36) Liu CJ, Jin JD, Lv TD, Wu ZZ, Ha XQ. Keratinocyte growth factor gene therapy ameliorates ulcerative colitis in rats. World journal of gastroenterology: WJG. 2011;17(21):2632-40. doi: 10.3748/wjg.v17.i21.2632.

37) D'Amici S, Ceccarelli S, Vescarelli E, Romano F, Frati L, Marchese C, Angeloni A. TNFalpha modulates Fibroblast Growth Factor Receptor 2 gene expression through the pRB/E2F1 pathway: identification of a non-canonical E2F binding motif. PloS one. 2013;8(4):e61491. doi: 10.1371/journal.pone.0061491.